# Global Coastal Characteristics (GCC): A global dataset of geophysical, hydrodynamic, and socioeconomic coastal indicators

Panagiotis Athanasiou[1], Ap van Dongeren[1,2], Maarten Pronk[1], Alessio Giardino[3], Michalis Vousdoukas[4], Roshanka Ranasinghe[2, 1, 5]

[1] Deltares, Delft, The Netherlands
[2] Department of Coastal and Urban Risk & Resilience, IHE Delft Institute for Water Education, Delft, The Netherlands
[3] Climate Change and Sustainable Development Department, Asian Development Bank, Manila, Philippines
[4] Department of Marine Sciences, University of the Aegean, Mitilene, Greece
[5] Water Engineering and Management, Faculty of Engineering Technology, University of Twente, Enschede, The Netherlands

*Correspondence to*: Panagiotis Athanasiou (Panos.Athanasiou@deltares.nl)

## Abstract

More than 10 percent of the world's population live in coastal areas that are less than 10 meters above sea level (also known as the low elevation coastal zone – LECZ). These areas are of major importance for local economy, transport and are home to some of the richest ecosystems. At the same time, they are quite susceptible to extreme storms and sea level rise. During the last few years numerous open access global datasets have been published, describing different aspects of the environment such as elevation, land-use, waves, water-levels and exposure. However, for coastal studies it is crucial that this information is available at specific coastal locations and, for regional studies or upscaling purposes, it is also important that data is provided in a spatially consistent manner. Here we create a Global database of Coastal Characteristics (GCC) with 80 indicators covering the geophysical, hydrometeorological and socioeconomic environment, at a high alongshore resolution of 1 km and provided at ~730,000 points along the global ice-free coastline. To achieve this, we use the latest freely available global datasets and a newly created global high-resolution transect system. The geophysical indicators include coastal slopes and elevation maxima, land-use, presence of vegetation or sandy beaches. The hydro-meteorological indicators involve water level, wave conditions and meteorological conditions (rain and temperature). Additionally, socioeconomic indices related to population, GDP and presence of critical infrastructure (roads, railways, ports and airports) are presented. While derived from existing global datasets, these indicators can be valuable for coastal screening studies, especially for data-poor locations.

## 1    Introduction

Between 750 million and 1.1 billion people live in low-lying coastal areas (MacManus et al., 2021) which is expected to increase in the future (Neumann et al., 2015) . This highlights the considerable economic significance of coastal regions, which also represent some of the planet's most valuable ecosystems (Paprotny et al., 2021). As a dynamic interface between land and sea, coastal zones are exposed to various environmental forces (including e.g., waves, currents, and wind) and socio-economic

pressures (urbanization, tourism and industry), all of which are continually changing at different spatio-temporal scales. Moreover, sea level rise driven by climate change (IPCC, 2021) is expected to lead to permanent submergence of low-lying coastal areas, more severe and frequent episodic flooding, shoreline retreat, wetland loss and freshwater degradation due to saltwater intrusion (IPCC, 2021; Ranasinghe et al., 2021; Glavovic et al., 2022).

5   The global coastline is quite diverse with various types of landforms including beaches, barrier islands, atolls, river deltas, cliffs, estuaries, and tidal flats, which can respond differently to marine forcing and support different land uses. These different landforms can be usually distinguished by differences in their geophysical and/or hydrometeorological characteristics. The shape of the coast, e.g., slope of the submerged or subaerial profile and average elevation, the presence of vegetation and generally the land cover, are just some of the characteristics that can have a strong influence in the response to increased water levels, flooding, or erosion. On the other hand, offshore marine conditions including wave climate and hydrodynamics are important for the coastal response to marine forcing, while meteorological conditions like temperature and rainfall, can be associated with local vegetation and geology. Additionally, information on the local presence of population, assets and infrastructure is critical for coastal impact and risk assessments and identifying vulnerable areas.

Due to the ever-increasing Earth observations and computational power, large-scale coastal impact assessments have become more frequent in literature during the past years. One of the first attempts of a coastal vulnerability assessment at the global scale, was the DIVA tool (Vafeidis et al., 2008; Hinkel and Klein, 2009), which although state-of-the-art at the time of its development, used predefined segments at a relatively coarse resolution (Wolff et al., 2016; Athanasiou, 2022), and was based on lower resolution and accuracy datasets, available at the time. Satellite imagery has been used at the global scale as well, to get insights of the global dynamics of shoreline change at sandy beaches (Luijendijk et al., 2018); global land erosion/accretion trends (Mentaschi et al., 2018); shoreline change due to El Niño along Pacific coastlines (Vos et al., 2023) and at the global scale (Almar et al., 2023); identifying coastal vegetation (van Zelst et al., 2021); or deriving beach face slopes for Australia (Vos et al., 2022). Moreover, in Young et al. (2019), the global distribution of coastal cliffs was assessed using the SRTMv3 global digital elevation model (DEM). Athanasiou et al. (2019), used a combination of the global DEM MERIT and the bathymetric map of GEBCO, to estimate nearshore slopes, while in Vousdoukas et al. (2020b), future shoreline change was assessed at sandy beaches globally. In Almar et al. (2021), coastal overtopping was assessed globally, using slope and coastal maximum data derived from the global DEM ALOS, but using a relative coarse grid along the global coastline. Additionally, large scale coastal flooding has been assessed at the European (Vousdoukas et al., 2018, 2020a) and the global scale (Kirezci et al., 2020; Tiggeloven et al., 2020) using extreme water level boundary conditions and global DEMs. In Hulskamp et al., 2023 a combination of satellite imagery and geophysical data at a transect level was used to classify the global coastline to typologies, and the assess the dynamics of muddy coasts. The aforementioned studies are those to assess coastal impacts at large spatial scales using different data, grid resolutions and approaches, highlighting the need for consistent coastal data at the global scale.

During the last few years, new open access global datasets are becoming available. The latest freely available global DEM with a high resolution of 1 arcsecond (~ 30 m at the equator) is the CopernicusDEM (European Space Agency and Airbus, 2022). However, this type of DEM captures the surface rather than the terrain elevation, making them inaccurate in vegetated and urban environments. In an attempt to correct for these effects, new DEMs have been produced correcting the elevations in

vegetated and urbanized areas, such as FABDEM (Hawker et al., 2022), which is however not in the public domain, and more recently the open access DiluviumDEM (Dusseau et al., 2023) and DeltaDTM (Pronk et al., 2024). Moreover, the European Space Agency (ESA) produced a new land-cover dataset at a high resolution of 10 m (Zanaga et al., 2021), while continuously updated global reanalysis products like ERA5 (Hersbach et al., 2020) provide an unprecedented representation of the historical climate conditions globally, and global hydrodynamic modelling (Muis et al., 2023) provide information on global water levels

statistics. Additionally, global population maps with a high resolution of 100 m from WorldPop (Bondarenko et al., 2020), and global mapping of infrastructure using crowd sourcing techniques (OpenStreetMap contributors, 2017), provides an enormous wealth of information on the exposed population and assets.

In this work, we use all these recent developments in global data to extract a large set of indicators at the coast. We present a compiled global database of coastal characteristics (Global Coastal Characteristics – GCC) based on indicators of the

geophysical, hydrometeorological and socioeconomic environment (Figure 1). The geophysical indicators are related to the profile shape characteristics (e.g., slope, elevation maxima etc.), presence of sandy beach or vegetation and land cover (e.g., build-up areas, forest etc.). The hydrodynamic indicators include offshore water level and wave conditions, and meteorological conditions (rain and temperature). Furthermore, socioeconomic indicators related to population, GDP and presence of critical infrastructure (roads, railways, ports and airports) are also included. For the extraction of all these different types of indicators,

we use available open access global datasets and generated a global transect system at a 1 km interval, resulting in ~730,000 points along the global ice-free coastline. To our knowledge, this is the first time that such a high number of diverse indicators, covering the spectrum between the geophysical characteristics, offshore marine environment, and socioeconomic conditions, are brought together on a common high-resolution grid, using state-of-the-art open access datasets. While based on global data, we believe that this dataset can be valuable for initial case study descriptions in data poor environments, quick coastal

screening studies or for broad coastal classification purposes. However, it should be noted that, given the resolution of the parent data sets and underlying assumptions, the database presented here is not meant to be used for detailed local scale modelling or assessments. Therefore, the use of this dataset for this kind of assessments, for example at data poor locations, should necessarily take into account the underlying limitations and uncertainties.

The paper is structured as follows: the underlying data and methodologies used to derive the indicators are described in Section

2. The results, including the definitions of all extracted indicators, and a set of global statistics and maps are presented in Section 3. The paper concludes with a discussion on the limitations associated with the presented dataset and an overview of its potential applications, with an example of a global classification of coasts using unsupervised machine learning.

## 2 Materials and methods

The prerequisites for all data sets used here to extract coastal indicators was that the datasets were recent open-access and had a global coverage. While we recognize that there are more potential candidates for each type of dataset, we based our selection on resolution, accuracy and coverage. These indicators were largely connected with the date of publication. In general, we used the latest datasets that were available at the time of this study. An overview of the data sets used, for which purpose they were used and their references are given in Table 1. For example, for land cover information we used the ESA World Cover map, since it had the highest resolution of global land cover products available at the time and a high accuracy of 77% (Zanaga et al., 2021). A general overview of the work-flow followed to create the GCC database is presented in Figure 2. The generation of a global transect system and the methods to extract the various indicators from the datasets, based on these transects, are presented in more detail in the next sections.

### 2.1 Transect system

A global transect system was generated using an along-coast spacing of 1 km in the zoom level 8 generalized version of the Open Street Maps (OSM) coastline vector (Generalized coastlines, 2016), which is a simplified version that removes fine details of the coastline (e.g., rocky outcrops) and has been previously used for other global transect systems (Luijendijk et al., 2018; Athanasiou et al., 2019). The choice of 1 km spacing was based on a balance between capturing information from high-resolution datasets (e.g., DEM and land use), while keeping the number of transects manageable from a data storage and computational point of view. This alongshore resolution offers a far better representation of the alongshore variability in comparison to previous studies where coarser resolution or segments approach were used (Wolff et al., 2016; Almar et al., 2021). Moreover, the transect system allows for future updates of the indicators based on new datasets that become available. The process adopted for creating the transects was as follows: First, a simple smoothing procedure was applied on the coastline to ensure that there were no sharp corners that could lead to misalignments of the transects. The smoothing procedure was conducted using Chaiken's algorithm as implemented in QGIS (QGIS Development Team, 2023) using a 0.25 offset, 180 maximum node angle and 5 iterations. Since our analysis focused on the ice-free coastal areas, coastlines in the South Pole (Antarctica) and at the North Pole (Greenland, and parts of Canada and Russia) were excluded from the analysis following Luijendijk et al. (2018). Then using the spacing of 1 km, shore normal transects were created by using the local orientation of the coast at each 1 km interval. This resulted in 728,088 cross-shore transects globally (Figure 3). The transects extended 4 km in both the landward and seaward directions from their centroid, which was defined at the OSM location (which, since we use a generalized coastline, is not necessarily the shoreline location). A total profile length of 8 km was chosen after testing different lengths to ensure a good coverage of the coastal zone, while avoiding unnecessarily high data-storage (Athanasiou et al., 2019). While the 4 km length landwards might not always cover the full extents of coastal floodplains at flat coastal areas, this choice was deemed appropriate, since the focus of the derived dataset is to capture the coastal characteristics close to the

coastline and not to perform coastal inundation modelling. The longitude and latitude of each centroid was extracted for each transect along with the transect orientation.

## 2.2 Raster data extraction

Various raster-based datasets were used to extract elevation, land cover and population information at each transect. This was performed with one of the following approaches, as appropriate: 1) by extracting a cross-shore profile of values along each transect or 2) by performing zonal statistics in a buffer area around the landward part of each transect (Figure 4).

Profiles were extracted from the topography, bathymetry, land-mask, land cover and water occurrence rasters (Table 1). Along each transect, two 1D grids with different cross-shore resolutions were defined; a coarse grid with a spacing of 25 m and a finer grid with a spacing of 10 m. This was done to ensure that raster datasets with different resolutions will be captured in detail, while also reducing the data-storage. The profiles of topography (both for CopernicusDEM and DeltaDTM) and bathymetry (GEBCO) were extracted along the coarse 25 m grid. The land mask (CopernicusDEM) and water occurrence (JRC) profiles were extracted on the same coarse grid. Finally, the land cover (ESA WorldCover) profile was extracted on the finer 10 m grid. This process resulted in five profiles with 321 points with values from the 1) CopernicusDEM elevation, 2) DeltaDTM elevation, 3) GEBCO bathymetry, 4) CopernicusDEM water-mask, and 5) JRC water occurrence; and one profile with 801 points with values from the ESA World Cover land-cover for each transect. All these profiles were later used to derived transect-wise indicators (see Section 2.4).

Zonal statistics were performed on the land cover (ESA WorldCover) and population (WorldPop) rasters using buffer zones of 500 m width around the transects (Figure 4). It should be noted that these buffer zones can be overlapping or not cover the whole 4 km (sub-aerial) zone, depending on the orientation of the coastline (see Section 4 for more details). For the land cover, the occurrence of each land cover class in the buffer zone was calculated for each transect (for the available land-cover classes in ESA World Cover please see Table 1). For the population, the total population in the buffer zone was calculated per transect and in addition, the number of people located below specific elevation thresholds (1, 5 and 10 m above MSL) was estimated using the DeltaDTM topography raster. To achieve this, the elevation raster was re-sampled to the population raster grid using averaging, and then masks were created when the elevation cells were below the respective elevation thresholds.

## 2.3 Vector data extraction

Various GIS methods were used to extract data from other datasets (Table 1) that were in points or lines (vector) format (Figure 5).

To assess the typology of the coast, the global coastal transect type classification from Hulskamp et al. (2023) was used. In that study, a different transect system (with 500 m alongshore spacing) was employed to classify coastal locations as sandy, muddy, rocky, vegetated or other type, using machine learning methods with satellite imagery and other geophysical indicators

as input. To classify our transects by a coastal type, proximity analysis was used (see Figure 5a) to determine the presence of points from Hulskamp et al. (2023) in a 1 km buffer zone around the centroid of our transects. If more than one points were present in the buffer zone, the closest point was selected. We followed the same approach to assess whether there is coastal vegetation that can protect the coast (i.e. mangroves or salt marshes) at a transect by using the data points from van Zelst et al.

(2021). A 1 km proximity analysis was also used to assess whether there are ports (using the World Port Index) or airports (using the Global Airport Database) in the vicinity of the transects.

To assess whether there are roads or railways at the transects, we performed an intersection analysis (see Figure 5b), where we determined whether a transect is intersecting with any line that represents a road (using the gRoads_v1 dataset) or a railway (using the Global Railways dataset).

Finally, to sample indicators describing continuous variables from point vector datasets, we used an inverse distance interpolation using the two closest points to each transect (see Figure 5c). These indicators were associated with wave and meteorological data from ERA5, water level data from the Global Tide and Surge Model (GTSM_v3.0) and gridded GDP data from Kummu et al. (2018). Only the two closest points that were in a zone of 100 km around each transect were used in the process, to avoid sampling of points that were far away and thus not descriptive of the transect's conditions. Since the gridded

data in Kummu et al. (2018) do not offer a full coverage globally (e.g., Pacific SIDS), we additionally extracted GDP/capita using a country identifier for each transect and the country GDP/capita information from the World Bank (2022).

For characterizing the wave conditions at each transect, the ERA5 dataset was used. More specifically, the hourly time-series between 1979-2019 for the significant height of combined wind waves and swell (swh), the peak wave period (pp1d) and the mean wave direction (mwd) were used to extract specific indicators at the ERA5 grid locations (for more information on the

20 parameters see (Hersbach et al., 2020)). These indicators included the $50^{th}$ and $95^{th}$ percentiles of the swh and pp1d, as indicators of the wave height and period during average and more extreme conditions. Additionally, the average mwd when the swh was larger than the 95th percentile of swh, was estimated as an indicator of the mean wave direction during extreme waves. The ERA5 wave data are available only at offshore locations due the coarse spatial resolution of the wave grid (~30 km). These conditions are not descriptive of the local wave environment at each transect. However, transforming 30 years of

25 wave time-series to ~1 million locations globally was out of the scope of the present study. To this end, all the wave indicators at each transect were sampled from the two closest ERA5 offshore points, with a 100 km maximum distance. The ERA5 dataset was additionally used to extract indicators for mean daily temperature (t2m) and total daily precipitation (tp), for both of which the 50th and 95th percentiles were extracted from the hourly time-series between 1979-2019. Water level indicators were extracted from GTSM_v3.0, including indicators related to the tidal level (TL), storm surge level (SSL) and the total

30 water level (TWL, including the mean sea level, tide and storm surge). These included the mean higher high water (mhhw) and mean lower low water (mllw), $50^{th}$ and $95^{th}$ percentiles of SSL, and the SSL and TWL return values for 1, 2, 5, 10, 25, 50, 100 years (for more information on the generation of these indicators in GTSM_v3.0 please see Muis et al., 2023).

## 2.4    Profile indicators extraction

To create a continuous elevation profile including both the submerged and subaerial parts of the coastal profile, the topographic and bathymetric profiles were merged. Two elevation profiles were created per transect, one using the CopernicusDEM and one using the DeltaDTM as the topography source. The following steps were followed : 1) The CopernicusDEM mask profile was used to define the land cells, 2) The CopernicusDEM or DeltaDTM topography profile values were used for the land cell elevations after they were transformed from "m above geoid" to "m above MSL" using the mean dynamic topography map (DTU10_MDT), the value of which was saved in the database as well, 3) Then the finer resolution "open water" class for the ESA World Cover raster was used to define the sea cells, 4) For the sea cells, the bathymetric raster values were used only when they were below MSL. If there were points that ESA World Cover was showing as land but the CopernicusDEM mask was indicating as water, an elevation of 0.2 m was used. Similarly, at the locations where the ESA World Cover was showing as water but GEBCO was showing as land, an elevation of -0.2 m was used. This was performed to ensure that at least the cross-shore location of the shoreline was based on the finer resolution of the ESA World Cover.

For each of the merged elevation profiles a set of locations that define specific areas of the profile were identified (Figure 6). The Depth of Closure (DoC) describes the depth seaward of which there is no significant change in bottom elevation at a specific timescale and is determined by the wave statistics (Hallermeier, 1978). In Athanasiou et al. (2019), an offshore wave reanalysis was used to estimate the DoC along the global coastline, with a temporal scale of 34 years following Nicholls et al. (1998). Here, the DoC at each transect was estimated by applying an inverse distance interpolation using the DoC values of the two closest offshore points from Athanasiou et al. (2019). A buffer zone of 150 km around each transect was used to sample the offshore points, to avoid values that are non-representative of the offshore wave environments of the transects being ascribed to them. While the depth of closure is a concept only valid for sandy coasts, here it is used broadly for all types of coasts to have an offshore limit for the profile shape calculations. In case there was no offshore point for DoC in the proximity, a default value of 10 m was used, and a warning was transcribed. The actual location of the DoC was calculated with respect to the mean lower low water, as extracted at the transect location from GTSM_v3.0 (see previous section). In case the subaqueous profile had more than one intersection with the DoC elevation, the most landward location was selected.

The location of the shoreline was identified as the most seaward location where the elevation profile crossed the value of 0 m, in a 1 km window around the centroid of the transect (which is the location of the smoothed OSM coastline). Then two different coastal maxima were identified landwards of the shoreline using two different methods. The coastal max (first peak) was identified by finding the first elevation peak landwards of the shoreline (Almar et al., 2021). The coastal max (max peak 1km) was identified by finding the maximum elevation peak in a 1 km window landwards of the shoreline. These two coastal maxima can act as a proxy of the local flood protection level. For both coastal maxima, the cross-shore location and the land cover type were extracted as well. The area landwards of the coastal max (first peak) was defined as the hinterland, for which the average hinterland elevation (he) and elevation variance (ev) were extracted.

The area between the depth of closure and the shoreline was defined as the nearshore (Athanasiou et al., 2019) and the nearshore slope (ns) was calculated as:

$$ns = \frac{z_{shoreline} - z_{DoC}}{x_{shoreline} - x_{DoC}} \tag{1}$$

The area between the shoreline and the coastal max (first peak) was defined as the backshore area and the backshore slope (bs) was calculated as:

$$bs = \frac{z_{coastal\_max\_first\_peak} - z_{shoreline}}{x_{coastal\_max\_first\_peak} - x_{shoreline}} \tag{2}$$

The coastal slope (cs) was calculated as well, defined as the slope between the depth of closure and the Coastal max (first peak). The coastal slope was calculated as:

$$cs = \frac{z_{coastal\_max\_first\_peak} - z_{DoC}}{x_{coastal\_max\_first\_peak} - x_{DoC}} \tag{3}$$

All the previously described indicators that included an elevation maxima in their calculation, were extracted twice for elevation profiles derived from both the CopernicusDEM and DeltaDTM to ensure that in case of potential overcorrections of the DeltaDTM, the CopernicusDEM derived values are available (see Table 2 and Section 4.1). Additionally, a warning/error

flag was assigned to each transect based on the calculations explained above. These flags were: 0 for No errors/warnings , 1 when a shoreline point could not be found, 2 for when the DoC value was not available for that transect and the -10 m was used, 3 for when the DoC was deeper than the deepest profile point (which was used for the calculation), 4 for when Coastal Max (first peak) could not be found and 5 for when the nearshore slope was steeper than 1:5 and the transect was indicated as sandy.

Finally, the width of the transition zone was extracted as the cross-shore distance between the points around the shoreline with a 5 and 95 % water occurrence as indicated in the JRC water occurrence map. This indicator combines information on the local tidal range and slope of the intertidal zone, but care should be taken with its interpretation since it could be affected by natural morphodynamics or human-induced changes.

## 3    Results

### 3.1    Geophysical indicators

A list of all the geophysical indicators extracted along with a description of each indicator and its label used in the GCC database is given in Table 2. As explained in the previous section the indicators that involved estimation of elevation maxima were extracted for both CopernicusDEM (copdem) and DeltaDTM (DeltaDTM). As an example, a global map of the coastal

max (max peak 1 km) using DeltaDTM is presented in Figure 7, along with some global statistics. A median coastal max of 13.64 m above MSL is found globally, with quite distinct spatial differences globally relating to the shape of the coast. The mode of the coastal max globally is between 3 and 4 m, while the 5th and 95th percentiles are 0.9 m and 260 m respectively. The peak in the distribution just above 100 m can be explained by the correction algorithm of DeltaDTM, which is limited above 100 m. For ~12% of the global transects, a coastal max with this method could not be calculated, since there was no elevation peak in the 1 km zone.

## 3.2 Hydrodynamic indicators

A list of all the hydrodynamic indicators extracted along with a description of each indicator and its label used in the GCC database is given in Table 3. As an example, a global map of the Average MWD of Hs ≥ Hs_p95 is presented in Figure 8, along with some global statistics. A median $H_s$ (95th percentile) of 210 deg N is found globally, with quite distinct spatial differences globally relating to global wind-patterns and the orientation of the coastline.

## 3.3 Socioeconomic indicators

A list of all the socio-economic indicators extracted along with a description of each indicator and its label used in the GCC database is given in Table 4. As an example, a global map of the population at elevation < 10 m is presented in Figure 9, along with some global statistics. More than 50% of the transects have 0 population at elevation < 10 m, while for the transects that have at least one person, the median value is ~200 people, with a distribution that is close to a normal distribution in the logarithmic scale. The 25th and 75th percentile values are 40 and 1000 people respectively for the transects with at least one person.

## 4 Discussion

### 4.1 Limitations

Bringing together a wide range of different datasets, describing geophysical, hydrometeorological, ecological and socio-economic characteristics, under a common, relatively high-resolution spatial grid provides an unprecedented holistic view of the global coastline. Nevertheless, while global datasets provide an (almost) global coverage, they arguably suffer from numerous shortcomings related to accuracy and resolution. Therefore, the developed indicators of the GCC dataset should not be seen as a replacement for local-scale studies or data-collection and monitoring campaigns and should not be directly employed for very local scale efforts such as, for e.g., designing coastal adaptation measures or informing decision making. On the other hand, the dataset presented herein can be a valuable tool for getting initial insights at data-poor environments; coastal screening studies for identifying hazard/impact hotspots where more detailed assessment should take place (van Dongeren et al., 2018); or coastal assessments at regional, continental or global scales, where aggregated statistics are suitable.

Since all our indicators are based on open access datasets, the methodologies used for their generation, detailed information on their accuracy, and underlying assumptions can be found in their respective references (Table 1). Therefore, here we only briefly discuss the main limitations in the original datasets or those introduced in our data processing to derive the indicators.

When combining topographic and bathymetric elevation data from different sources and with different resolutions, the transition zone between sea and land can be a challenging area to correctly resolve. Here, since we focused on the derivation of indicator, as opposed to reproducing accurate depths and elevations in this transition area, we employed the land-cover map to distinguish between land and sea (see Section 2.4). In this way we have more confidence in the cross-shore location of the shoreline, which is used for the extraction of different slope indicators. An improvement would be to explore how EO derived intertidal bathymetry could be used to obtain better estimates in this transition zone (Mason et al., 1995).

Moreover, as mentioned in the introduction, global DEMs like CopernicusDEM, suffer from biases in areas with high vegetation and buildings. Corrected DEMs, like FABDEM and DeltaDTM account for these biases and produce elevation maps far closer to the actual terrain elevation. However, it has been observed, that due to the underlying algorithms applied for this correction, they can in some cases overcorrect narrow features like dunes and dikes, reducing their elevations. For this type of locations, CopernicusDEM was found to perform better in capturing the elevation (when compared with local in-situ data). Thus, in the GCC dataset we have provided all the elevation related parameters, both for CopernicusDEM and DeltaDTM. In areas with narrow, vegetation-less dunes, CopernicusDEM will capture coastal maxima more accurately, while in areas with coastal vegetation, DeltaDTM will produce better results. For indicators such as the mean hinterland elevation and population below given elevations, which cover larger areas rather than narrow features, DeltaDTM derived indicators are likely to be more accurate, that is why we provide a single value.

The population and land-cover indicators that were extracted from the buffer zones around the transects (see Figure 4), give an indication of the local transect conditions, but do not necessarily depict spatially representative values. For example, when the coast has a convex shape (e.g., at a peninsula) the transects and thus buffer zones can overlap, meaning that the same cell of population or land-cover from the initial raster datasets can be counted in more than one transect. On the other hand, when the coast has a concave shape (e.g., at an embayment), the buffer zones will not cover the entire area 4 km landwards of the shoreline, as the transects will diverge over land. Furthermore, at narrow islands too the transects and their buffer zones may overlap (see Figure 3). The purpose of the population and land-cover indicators is to provide an indication of the exposure and the socioeconomic characteristics of the area around the transect. To that end, these indicators should not be used to calculate aggregated summed values e.g., to estimate the total population near the coastline.

While we provide indicators of population counts at different elevation thresholds, it should be noted that the uncertainties in the derived values can be high, since they are dependent on the accuracy of the global elevation model used. In the case of the present study, we used DeltaDTM which has a vertical mean absolute error of 0.45 m overall (Pronk et al., 2024), which gave us confidence on the derived indicators giving a good approximation of the exposed population.

Another limitation of the GCC dataset is with respect to the wave related indicators, which are sampled at the local transect level from ERA5 offshore points around a 100 km buffer zone with a simple inverse distance interpolation. An offshore to nearshore wave transformation for almost one million locations globally is beyond the scope of this global dataset. However, this methodological constraint can lead to significant differences between the wave indicators extracted and the actual wave conditions at each transect, where the resolution of the ERA5 wave grid cannot represent the shape of the coastline and wave refraction/diffraction effects on nearshore wave conditions.

It is important to note that our indicators in the GCC dataset set presents a static image of the coast. However, a large part of the global coastline can be quite dynamic in response to hydrometeorological forcing and/or human developments. Using global-DEMs allows us to capture a moment in time of the shape of the coastline, which will produce inconsistencies in highly dynamic areas or fast developing areas. The same holds for the land-use data, which can have seasonal variations and more permanent changes especially in highly developed coastal zones. Latest developments using satellite imagery to produce near real time land-use maps (Brown et al., 2022) indicate that the dynamic component of land-use could potentially be taken into account, while in the future similar techniques could allow for better representation of dynamic elevations when the respective sensors and techniques develop further. Except from the data that are used to sample the indicators at the transect level, this dynamic component of the coast can affect the location and number of transects themselves, while in some cases inconsistencies between the timing of the shoreline and DEM data can result in non-representative indicators.

## 4.2    Potential applications

The GCC dataset can have numerous applications. For example, coastal screening tools using indices; like the coastal vulnerability index (CVI) have been commonly applied at large spatial scales for impact assessments (Rocha et al., 2023; Pantusa et al., 2022). These tools depend on various indices related to the geophysical shape, hydrodynamic forcing and exposure; and the indicators produced herein could directly be used for this purpose. Furthermore, these indicators could be useful for gaining initial insights for high-level studies aimed at planning future regional or country scale developments.

During the last decade or so various studies have assessed climate change impacts on coasts globally including sandy beach erosion (Athanasiou et al., 2019, 2020; Vousdoukas et al., 2020b) and coastal flooding (Kirezci et al., 2020; Tiggeloven et al., 2020; Almar et al., 2021; van Zelst et al., 2021). For this type of studies the produced indicators in GCC can be a valuable input in combinations with climate-change scenarios (IPCC, 2021), enabling the assessment of future coastal impacts in a globally consistent way. For example, the nearshore slope can be used for assessing wave transformation and sea level rise induced shoreline retreat, while the backshore slope can be employed for estimating wave run-up. Also, the coastal maxima and hinterland elevation indicators may be used as a representation of the coastal protection level, potentially replacing purely qualitative rules-based approaches (Scussolini et al., 2016).

The typology of the coast can dictate its vulnerability to environmental forcing and future climate change. For large scale assessments this classification is valuable to differentiate between different hazards and the relevant approaches that are needed to assess the impacts at different parts of the coast. Coastal classification relies on coastal indicators in order to group coastlines of a certain type together (Finkl, 2004; Rosendahl Appelquist and Halsnæs, 2015; Dang et al., 2020; Mao et al., 2022). As a

showcase of the potential use of the dataset presented here for coastal classification, we demonstrate the application of an unsupervised global classification based on a set of geophysical indicators, grouping the global coastline based on its geophysical shape. For this, we apply a K-Means clustering, following the approach in Athanasiou et al. (2021), using six geophysical indicators from the presented GCC database: 1) Coastal Max (max peak 1 km) , 2) Mean hinterland elevation , 3) Nearshore slope, 4) Backshore slope, 5) Open-water land cover class occurrence, and 6) Transition zone width. Since this is

not a dedicated global coastal classification analysis, but rather an indicative application of the presented dataset, we only use 9 clusters (Figure 10). This value can be quite low for a global clustering, but it allows for easy visualization of the results. Due to the choice of the low number of clusters, the intra-cluster variability of the elevation profiles is quite high (Figure 10c). This high variability can be identified by the large envelopes of the 5th-95th percentile range in the cross-shore elevation profiles for each cluster. However, there are distinct shapes between the different clusters. For example, clusters 3 and 7

represent coastlines with high coastal maxima, high hinterland elevation and steep slopes, which, as seen in the global map in Figure 10a can be connected with fjord areas (e.g. Norway and Chile). On the other hand, cluster 6 shows the highest occurrence of open-water landwards of the shoreline and generally low elevations, which can be connected with atoll islands (e.g. Pacific SIDS).

## 5    Data Availability

The GCC dataset can be accessed at Zenodo under https://zenodo.org/doi/10.5281/zenodo.8200199 (Athanasiou et al., 2024). The dataset is composed of three comma separated values (.csv) files with the global geophysical, hydrodynamic and socioeconomic indicators respectively and a metadata file describing all the indicators included in the csv files.

## 6    Conclusions

Here, we provide a 1 km along-coast resolution dataset of 80 coastal indicators describing the geophysical, hydrometeorogical,

and socio-economic conditions along the global ice-free coastline. We believe that our dataset can act as a significant asset for future global assessments of  coastal hazards and impacts, since it provides information in a consistent and therefore comparable manner. The dataset offers a collection of indicators that can be useful for coastal screening studies, quick overviews of coastal environment anywhere in the world, and for coastal classification. The indicators were extracted based on various open-source global datasets that were available at the time of undertaking this work, but can be updated using the

same workflow, whenever new datasets become available.

**Author contribution**

PA, AvD, AG, MV, and RR designed the study. PA carried out the data processing and analysis. MP created the corrected DEM used in the study and assisted with the DEM related parts. PA prepared the paper with contributions from all co-authors.

**Competing interests**

5 The authors declare that they have no conflict of interest.

**Disclaimer**

The views expressed in the article do not reflect the views of ADB.

**Acknowledgements**

RR is partly supported by the AXA Research Fund. PA and AvD were partly funded through a contract with the Joint Research 10 Centre of the European Commission.

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

**Tables**

Table 1: Summary of all the datasets used for the generation of the global coastal characteristics database.

| Dataset name | Type | Description | Purpose | Reference |
|---|---|---|---|---|
| **OSM coastline (zoom 8 level) 2016** | Vector (lines) | Generalized coastline from Open Street Maps | Describe the global location of the shoreline and create the transects. | (Generalized coastlines, 2016) |
| **CopernicusDEM (GLO-30)** | Raster | Global topographic raster dataset at ~30 m resolution | Extract the subaerial part (topography) of the elevation profiles and the profile of the water mask used in GLO-30. | (European Space Agency and Airbus, 2022) |
| **DeltaDTM** | Raster | Global coastal topographic raster dataset at ~30 m resolution | Extract the subaerial part (topography) of the elevation profiles. | (Pronk et al., 2024), unclipped version with values above 10 m +MSL |
| **GEBCO 2023 Grid** | Raster | Global bathymetric raster dataset at ~500 m resolution | Extract the submerged part (bathymetry) of the elevation profiles. | (GEBCO Bathymetric Compilation Group, 2023) |
| **DTU10_MDT** | Raster | Global Mean Dynamic Topography | Transform topographic elevation from the EGM2008 geoid reference to mean sea level (MSL) vertical reference. | (Knudsen and Anderson, 2013) |
| **ESA World Cover (v100)** | Raster | Global land cover class raster at ~10 m resolution with classes; 'Trees', 'Shrubland', 'Grassland', 'Cropland', 'Built-up', 'Barren / sparse vegetation', 'Snow and ice', 'Open water', 'Herbaceous wetland', 'Mangroves', 'Moss and lichen' | Classify coastal protection and extract the main land cover class for each transect. | (Zanaga et al., 2021) |
| **JRC water occurrence** | Raster | Global raster of water occurrence at ~ 30 m resolution | Get profile of water occurrence and extract transition zone width. | (Pekel et al., 2016) |
| **Global coastal type classification** | Vector (point) | Point data along the global coastline with coastal classes: Sandy, Muddy, Rocky, Vegetated or Other type | Identify the occurrence of a specific type of coast at a transect. | (Hulskamp et al., 2023) |

| | | | | |
|---|---|---|---|---|
| ***Global occurrence of saltmarshes and mangroves*** | Vector (point) | Geolocation of coastal segments with mangroves or saltmarshes | Identify the occurrence of vegetation at a transect. | (van Zelst et al., 2021) |
| ***ERA5*** | Raster (but points extracted for analysis) | Atmospheric, land and oceanic climate variables reanalysis | Extract the offshore significant wave height, peak period, mean wave direction, local temperature and precipitation indicators at each transect. | (Hersbach et al., 2020) |
| ***GTSM_v3.0*** | Vector (point) | Storm surge and tide reanalysis using a global hydrodynamic Delft3D Model | Extract tide, surge and total water level indicators at each transect. | (Muis et al., 2023) |
| ***WorldPop*** | Raster | Global population count per pixel at ~100 m resolution (Constrained individual countries 2020 UN adjusted) | Calculate the population indicators at each transect. | (Bondarenko et al., 2020) |
| ***World Bank GDP dataset*** | Tabular | GDP and GDP/capita for all countries | Characterize the GDP/capita per transect. | (World Bank, 2022) |
| ***Gridded global GDP*** | Raster (but points extracted for analysis) | GDP and GDP/capita for all countries | Characterize the GDP/capita per transect. | (Kummu et al., 2018) |
| ***gRoads_v1*** | Vector (lines) | Global Roads Inventory Network | Identify presence of roads at a transect. | (Meijer et al., 2018) |
| ***World Port Index*** | Vector (points) | Geolocation of major ports | Identify presence of a port in the proximity of a transect. | (National Geospatial-Intelligence Agency, 2019) |
| ***Global Airport Database*** | Vector (points) | Geolocation of major airports | Identify presence of a port in the proximity of a transect. | (The Global Airport Database, 2021) |
| ***Global Railways*** | Vector (lines) | Global railway lines vector from the World Food Programme | Identify presence of railways per transect. | (Global Railways, 2022) |

Table 2: Summary of all the geophysical parameters included in GCC.

| Parameter | Label | Units | Description |
|---|---|---|---|
| *Transect id* | id | - | Id of the transect in the form of BOX_{box corners}_{transect number in box} |
| *Longitude* | lon | degrees | Longitude of the centroid of the transect |
| *Latitude* | lat | degrees | Latitude of the centroid of the transect |
| *Transect angle* | angle | Degrees N | Angle of transect (from land to sea direction) |
| *Coastal Max (first peak)* | z_peak_first_{x} | m above MSL | First elevation peak found landwards of the shoreline using x: deltadtm (DOI: 10.1038/s41597-024-03091-9) or copdem (DOI: 10.5270/ESA-c5d3d65) |
| *Coastal Max (max peak 1 km)* | z_peak_max_1km_{x} | m above MSL | Maximum elevation peak found between the shoreline and 1 km landwards of the shoreline using x: deltadtm (DOI: 10.1038/s41597-024-03091-9) or copdem (DOI: 10.5270/ESA-c5d3d65) |
| *Cross-shore location of Coastal Max (first peak)* | x_peak_first_{x} | m | Cross-shore location of Coastal Max (first peak), with the transect's centroid as reference, using x: deltadtm (DOI: 10.1038/s41597-024-03091-9) or copdem (DOI: 10.5270/ESA-c5d3d65) |
| *Cross-shore location of Coastal Max (max peak 1 km)* | x_peak_max_1km_{x} | m | Cross-shore location of Coastal Max (max peak 1 km), with the transect's centroid as reference, using x: deltadtm (DOI: 10.1038/s41597-024-03091-9) or copdem (DOI: 10.5270/ESA-c5d3d65) |
| *Land cover class of Coastal Max (first peak)* | lu_peak_first_{x} | - | Land cover class of Coastal Max (first peak) as extracted from ESA Worldcover_v100 (DOI: 10.5281/zenodo.5571936), using x: deltadtm (DOI: 10.1038/s41597-024-03091-9) or copdem (DOI: 10.5270/ESA-c5d3d65) |
| *Land cover class of Coastal Max (max peak 1 km)* | lu_peak_max_1km_{x} | - | Land cover class of Coastal Max (max peak 1 km) as extracted from ESA Worldcover_v100 (DOI: 10.5281/zenodo.5571936), using x: deltadtm (DOI: 10.1038/s41597-024-03091-9) or copdem (DOI: 10.5270/ESA-c5d3d65) |
| *Mean hinterland elevation* | he | m above MSL | Mean elevation of the hinterland, landwards of Coastal Max (first peak) and excluding water areas, using deltadtm (DOI: 10.1038/s41597-024-03091-9) |
| *Variance of hinterland elevation* | ev | $m^2$ | Variance of the elevation of the hinterland, landwards of Coastal Max (first peak) and excluding water areas, using deltadtm (DOI: 10.1038/s41597-024-03091-9) |
| *Depth of Closure* | doc | m | Depth of closure found using Athanasiou et al. 2019 (https://doi.org/10.5194/essd-11-1515-2019). If distance of offshore location larger than 150 km a value of -10 m was assumed |

| | | | |
|---|---|---|---|
| *Depth of Closure used* | doc_used | m | Depth of closure used for slopes calculations in case the actual doc value was not available in the extracted profile |
| *Cross-shore location of Depth of Closure* | x_doc | m | Cross-shore location of Depth of Closure which was used in the profile, with the transect's centroid as reference |
| *Cross-shore location of shoreline* | x_shoreline | m | Cross-shore location of shoreline, with the transect's centroid as reference |
| *Nearshore slope* | ns | - | Nearshore slope calculated between the Depth of Closure and shoreline points |
| *Backshore slope* | bs_{x} | - | Backshore slope calculated between the shoreline and Coastal Max (first peak) points, using x: deltadtm (DOI: 10.1038/s41597-024-03091-9) or copdem (DOI: 10.5270/ESA-c5d3d65) |
| *Coastal slope* | cs_{x} | - | Coastal slope calculated between the Depth of Closure and Coastal Max (first peak) points), using x: deltadtm (DOI: 10.1038/s41597-024-03091-9) or copdem (DOI: 10.5270/ESA-c5d3d65) |
| *Error or warning flag* | extraction_flag | - | 0: No errors/warnings<br><br>1: Shoreline not found<br><br>2: Depth of Closure not available. Default -10 m used,<br><br>3: Depth of Closure is deeper than the deepest profile point (which is used for the calculation),<br><br>4: Coastal Max (first peak) could not be found<br><br>5: Nearshore slope is steeper than 1:5 and the transect is indicated as sandy<br><br>(multiple flags for the same transects are possible) |
| *Width of transition zone* | tr_zone_width | m | Width of transition zone defined as the two points around the shoreline with 5 and 95 % water occurrence as extracted from Pekel at al. 2016 (https://doi.org/10.1038/nature20584) |
| *x land cover class occurrence* | lu_{x} | % | Occurrence of land-cover class from the classes 'Trees', 'Shrubland', 'Grassland', 'Cropland', 'Built-up', 'Barren / sparse vegetation', 'Snow and ice', 'Open water', 'Herbaceous wetland', 'Mangroves', 'Moss and lichen' from the ESA Worldcover_v100 (DOI: 10.5281/zenodo.5571936). This is calculated in 500 m buffer zone around the landwards part of the transect |
| *Main land cover class* | class_most | - | Most encountered land cover class from ESA Worldcover_v100 (DOI: 10.5281/zenodo.5571936). This is calculated in 500 m buffer zone around the landwards part of the transect |
| *Main land cover class excluding open-water* | class_most_land | - | Most encountered land cover class from ESA Worldcover_v100 (DOI: 10.5281/zenodo.5571936) |

| | | | |
|---|---|---|---|
| | | | excluding the open-water class.This is calculated in 500 m buffer zone around the landwards part of the transect. |
| *Coastal type* | coast_type_flag | - | Occurrence of Sandy, Muddy, Rocky, Vegetated or Other type of coast in the proximity of the transect, based on Hulskamp et al. 2023 (DOI:10.1038/s41467-023-43819-6) |
| *Vegetation type* | veg_type | - | Occurrence of mangrove or saltmarsh vegetation in the proximity of the transect based on van Zelst et al. 2022 (DOI: 10.1038/s41467-021-26887-4) |
| *Mean Dynamic Topography* | mdt | m | Difference between MSL and the geoid extracted at the centroid of the transect with bilinear interpolation from DTU10_MDT (Knudsen and Andersen 2013)) |

Table 3: Summary of all the hydrodynamic parameters included in GCC.

| Parameter | Label | Units | Description |
|---|---|---|---|
| **Hs 50th percentile** | swh_p50 | m | 50th percentile of significant height of combined wind waves and swell (1979-2019) extracted from ERA5 (DOI: 10.1002/qj.3803) |
| **Hs 95th percentile** | swh_p95 | m | 95th percentile of significant height of combined wind waves and swell (1979-2019) extracted from ERA5 (DOI: 10.1002/qj.3803) |
| **Tp 50th percentile** | pp1d_p50 | s | 50th percentile of Peak wave period (1979-2019) extracted from ERA5 (DOI: 10.1002/qj.3803) |
| **Tp 95th percentile** | pp1d_p95 | s | 95th percentile of Peak wave period (1979-2019) extracted from ERA5 (DOI: 10.1002/qj.3803) |
| **Average MWD of Hs ≥ Hs_p95** | mwd_p95 | degrees N | Average Mean Wave Direction (MWD) relative to true North, when Hs ≥ Hs_p95 (1979-2019) extracted from ERA5 (DOI: 10.1002/qj.3803) |
| **Mean daily temperature 50th percentile** | t2m_p50 | degrees Celsius | 50th percentile of mean daily temperature (1979-2019) extracted from ERA5 (DOI: 10.1002/qj.3803) |
| **Mean daily temperature 95th percentile** | t2m_p95 | degrees Celsius | 95th percentile of mean daily temperature (1979-2019) extracted from ERA5 (DOI: 10.1002/qj.3803) |
| **Total daily precipitation 50th percentile** | tp_p50 | mm | 50th percentile of total daily precipitation (1979-2019) extracted from ERA5 (DOI: 10.1002/qj.3803) |
| **Total daily precipitation 95th percentile** | tp_p95 | mm | 95th percentile of total daily precipitation (1979-2019) extracted from ERA5 (DOI: 10.1002/qj.3803) |
| **Mean Higher High Water** | mhhw | m above MSL | Mean Higher High Water (MHHW) (1985-2014) extracted from GTSMv3.0 (DOI: 10.1029/2023EF003479) |
| **Mean Lower Low Water** | mllw | m above MSL | Mean Lower Low Water (MLLW) (1985-2014) extracted from GTSMv3.0 (DOI: 10.1029/2023EF003479) |
| **SSL 50th percentile** | ssl_p50 | m | 50th percentile of storm surge level (SSL) (1985-2014) extracted from GTSMv3.0 (DOI: 10.1029/2023EF003479) |
| **SSL 95th percentile** | ssl_p95 | m | 95th percentile of storm surge level (SSL) (1985-2014) extracted from GTSMv3.0 (DOI: 10.1029/2023EF003479) |
| **SSL (x years RP)** | ssl_rp{x}_mean | m | Storm surge level (SSL) with return period of x: 1, 2, 5, 10, 25, 50, 100 years, from a GPD fit (1985-2014) extracted from GTSMv3.0 (DOI: 10.1029/2023EF003479) |
| **TWL (x years RP)** | twl_rp{x}_mean | m above MSL | Total water level (TWL) with return period of x: 1, 2, 5, 10, 25, 50, 100 years, from a GPD fit (1985-2014) extracted from GTSMv3.0 (DOI: 10.1029/2023EF003479) |

Table 4: Summary of all the socioeconomic parameters included in GCC.

| Parameter | Label | Units | Description |
|---|---|---|---|
| *Country name* | country | - | Country standard name that the transects belongs to derived from the WorldPop (DOI: 10.5258/SOTON/WP00685) dataset |
| *Number of people* | pop_all | People | Total number of people located in 500 buffer area. This is calculated in 500 m buffer zone around the landwards part of the transect using the WorldPop (DOI: 10.5258/SOTON/WP00685) dataset |
| *Number of people below x elevation* | pop_{x}_m | People | Total number of people located in 500 buffer area and below an elevation x: 1, 5, and 10 m. This is calculated in 500 m buffer zone around the landwards part of the transect using the WorldPop (DOI: 10.5258/SOTON/WP00685) dataset |
| *Roads occurrence* | roads | - | Intersection of transect with roads (1) or not (0) using the gROADSv1 (DOI: 10.1088/1748-9326/aabd42) dataset |
| *Railways occurrence* | railways | - | Intersection of transect with railways (1) or not (0) using the WFP Global Railways dataset (https://geonode.wfp.org/layers/geonode%3Awld_trs_railways_wfp). |
| *Ports occurrence* | ports | - | Port occurrence (1) or not (0) at a radius of 1 km around the transect using the WPI 2019 (https://msi.nga.mil/Publications/WPI) dataset. |
| *Airports occurrence* | airports | - | Airports occurrence (1) or not (0) at a radius of 1 km around the transect using the Global Airport Database (https://www.partow.net/miscellaneous/airportdatabase/). |
| *GDP per capita PPP at 2015* | gdp_ppp_usd2017_2015 | USD (2017) | Gross Domestic Product (GDP) per capita based on purchasing power parity (PPP) at 2015, in USD 2017, extracted from the World Bank (https://data.worldbank.org/indicator/NY.GDP.MKTP.PP.KD) database based on the transect's country. |
| *Gridded GDP per capita PPP at 2015* | gdp_ppp_grid_2015 | USD (2011) | Gridded Gross Domestic Product (GDP) per capita based on purchasing power parity (PPP) at 2015, in USD 2011, extracted from Kummu et al. 2015 (DOI: 10.1038/sdata.2018.4). |

**Figures**

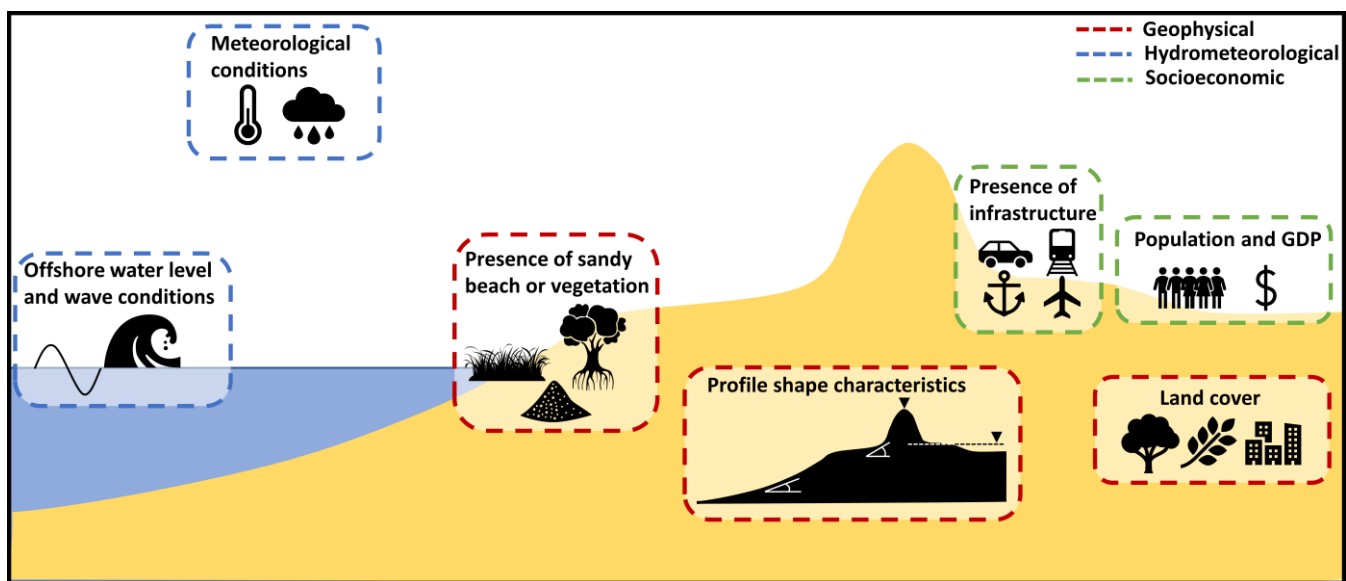

Figure 1: Overview of indicator groups extracted per transect globally. Indicators cover a wide range of coastal conditions, including the 1) geophysical (presence of sand or vegetation, profile shape characteristics and land cover), 2) hydrometeorological (offshore water level and wave conditions) and meteorological indicators and 3) socioeconomic environment (population, GDP and presence of critical infrastructure).

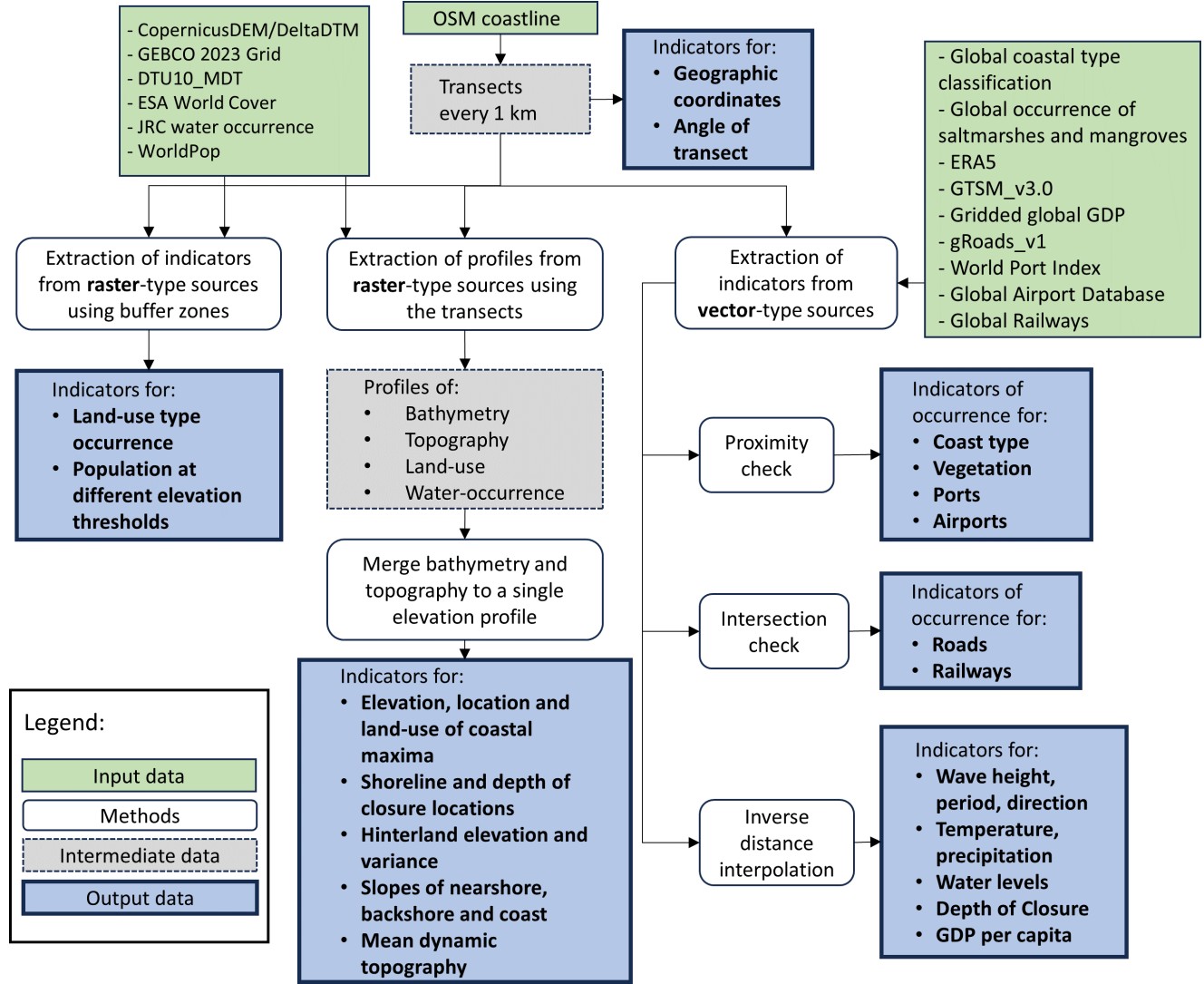

Figure 2: Flowchart of the work-flow to derive the indicators of the GCC database.

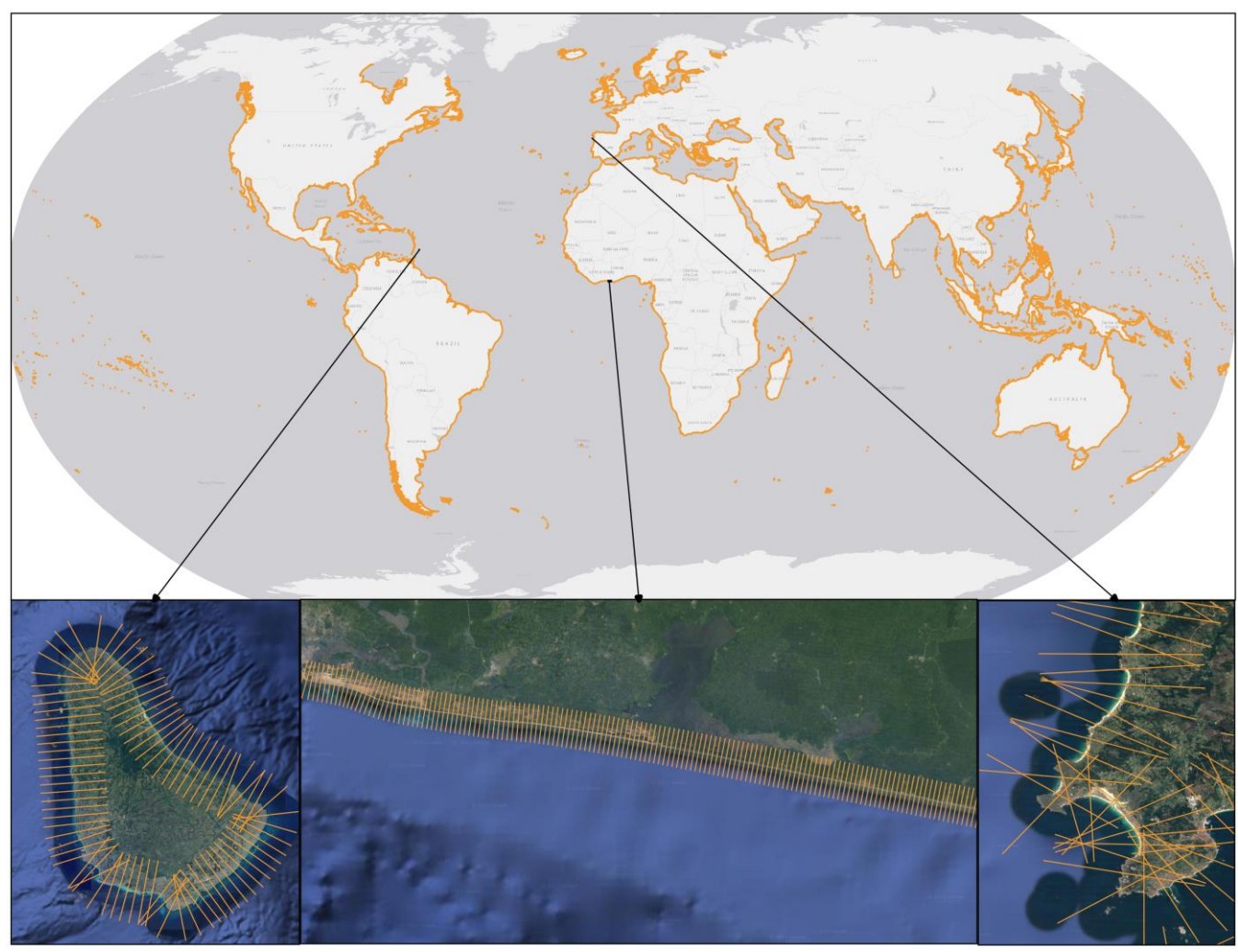

Figure 3: Top: Global coverage of the transect system. Bottom-left: Zoom in on Barbados. Bottom-centre: Zoom in on Ghana. Bottom-right: Zoom in on the west coast of Spain (Map data © Google Maps 2018).

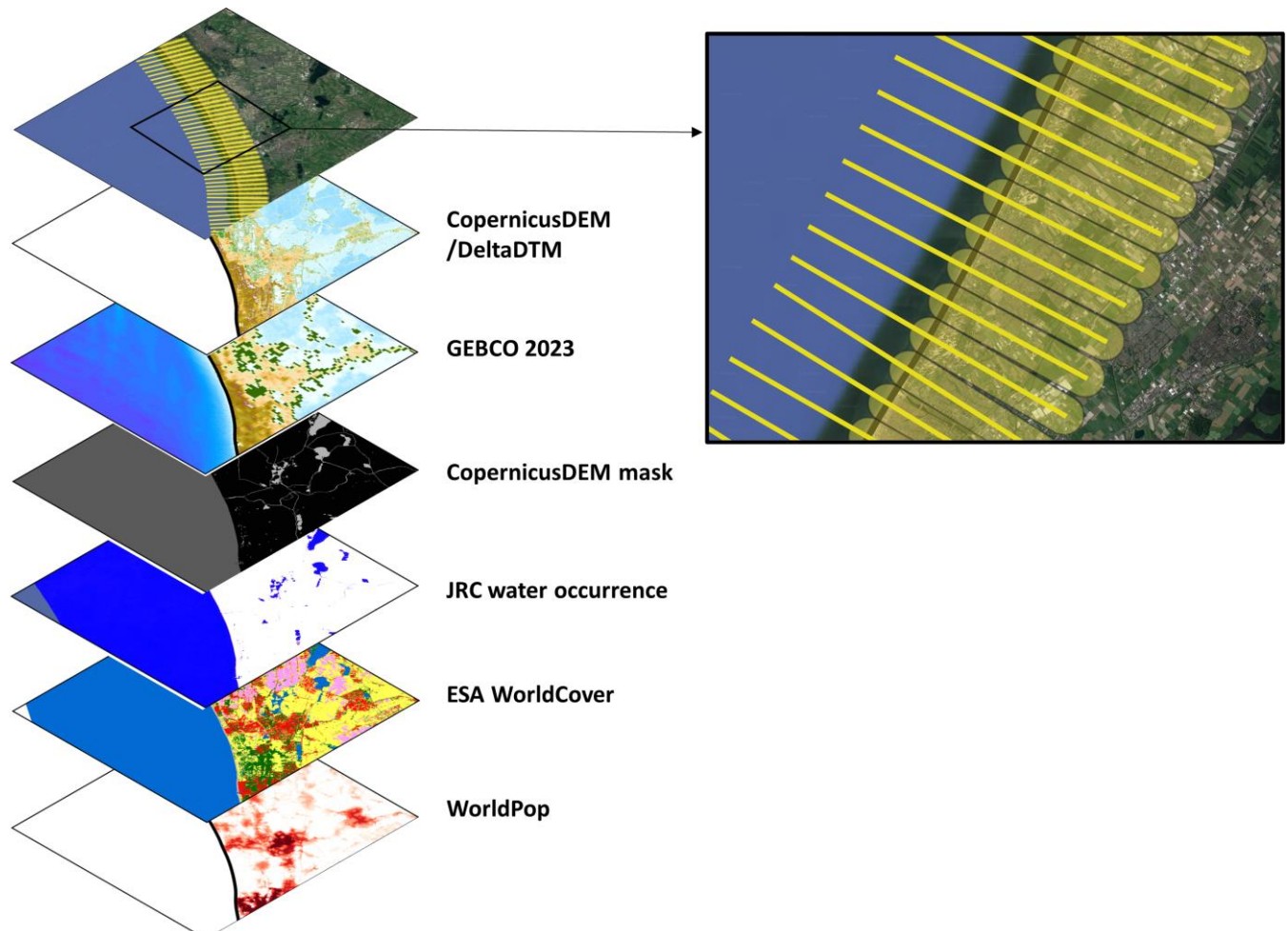

Figure 4: Raster datasets used for extracting transect information. The zoom-in shows the transects (yellow lines) along which the profiles were extracted, and the buffer zones (yellow areas) in which zonal statistics were performed (Map data © Google Maps 2018).

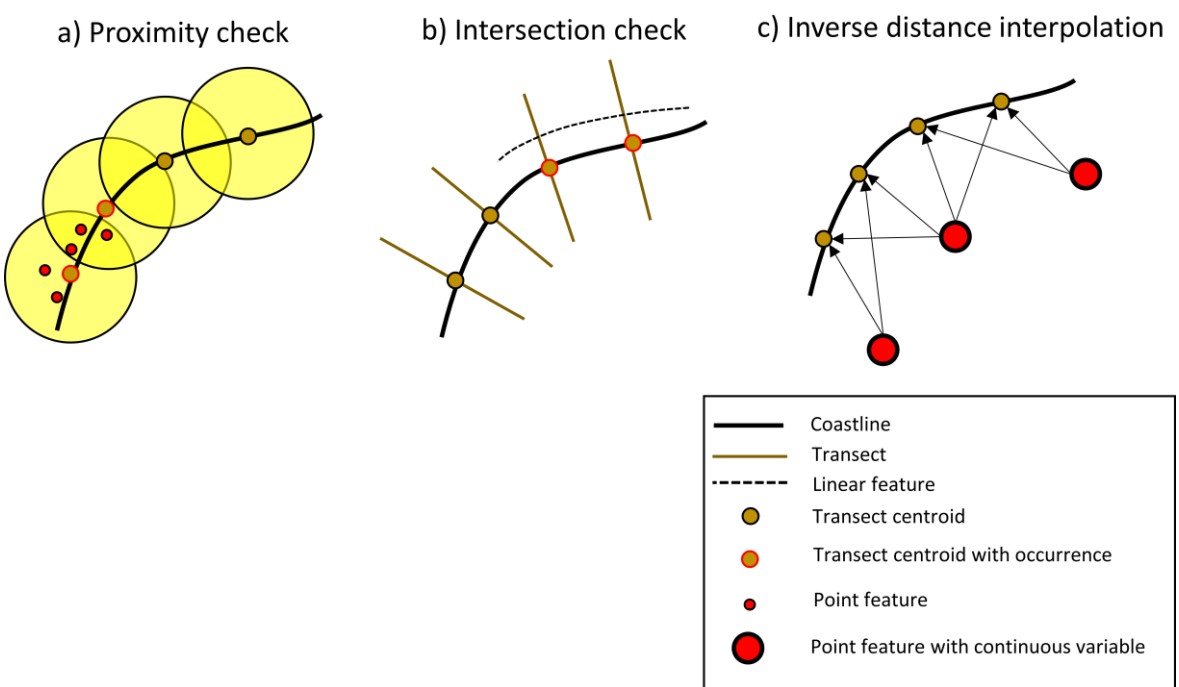

Figure 5: Schematic representation of the three methods followed to sample vector data to the transects for an indicative example coastline with four transects. a) Proximity check identifies if there are point features in a buffer zone around the centroid of a transect, b) Intersection check, identifies if the transect is intersecting a line feature and c) Inverse distance interpolation, uses the two closest points to get a distance weighted estimate at the transect.

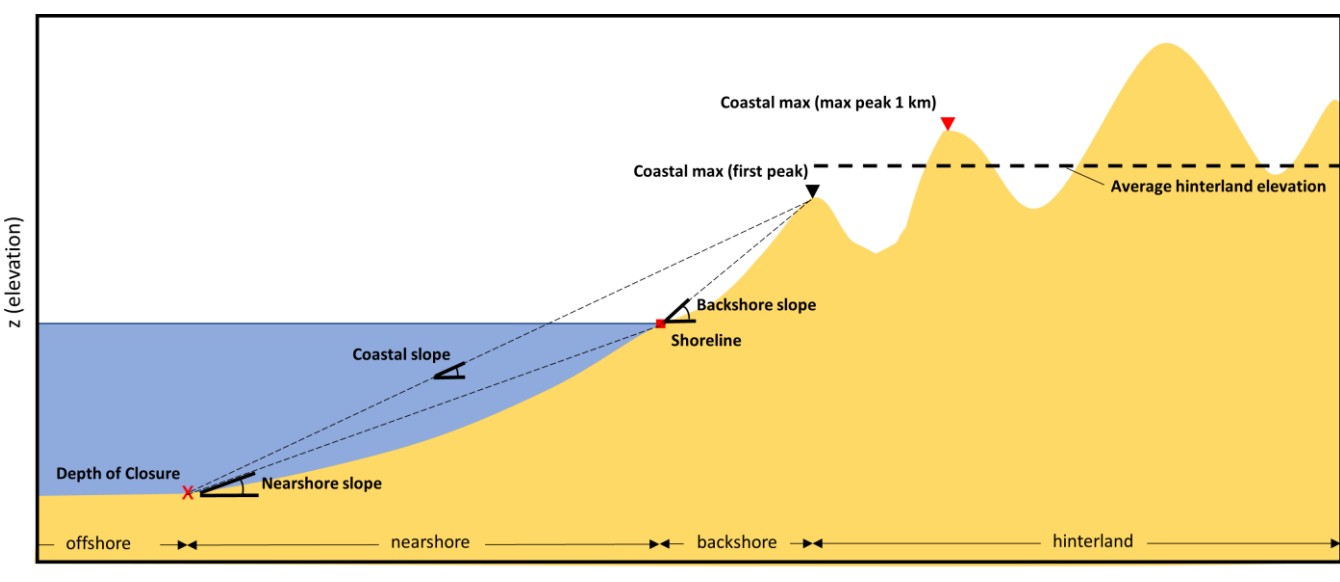

Figure 6: Schematic representation of the geophysical characteristics that are extracted per elevation profile

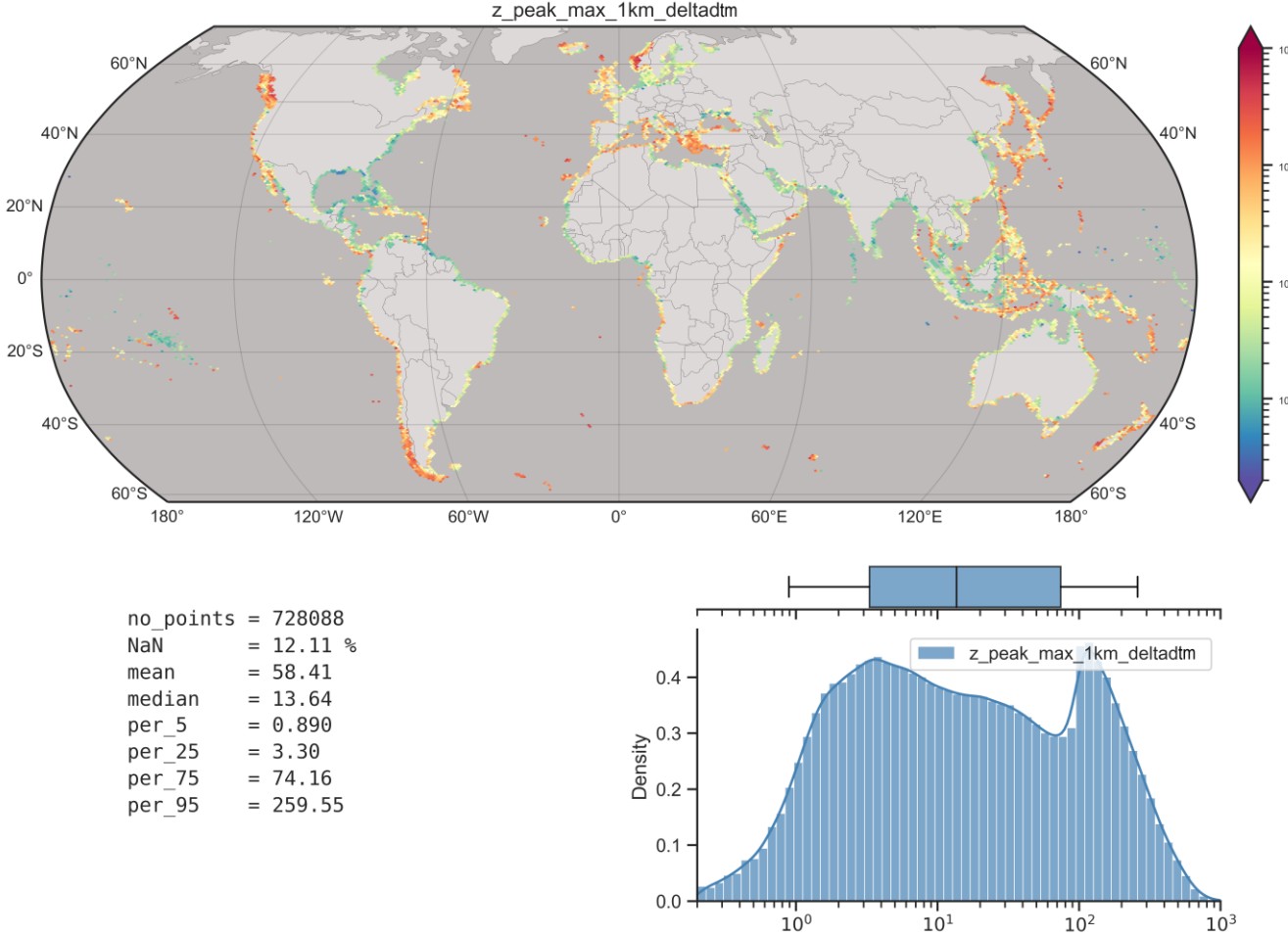

Figure 7: Global map and statistics of the Coastal max (max peak 1km) using DeltaDTM. Top: Global map showing the median values in hexagons with a size of ~1 degree. The color bar is given in a logarithmic scale. Bottom right: Histogram and boxplot (with the lines showing the 5th and 95 percentiles, the box the 25th and 75th percentiles, and the vertical line the median) of Coastal max (max peak 1km) at the transect level. The x is plotted in logarithmic scale. Bottom left: various global statistics.

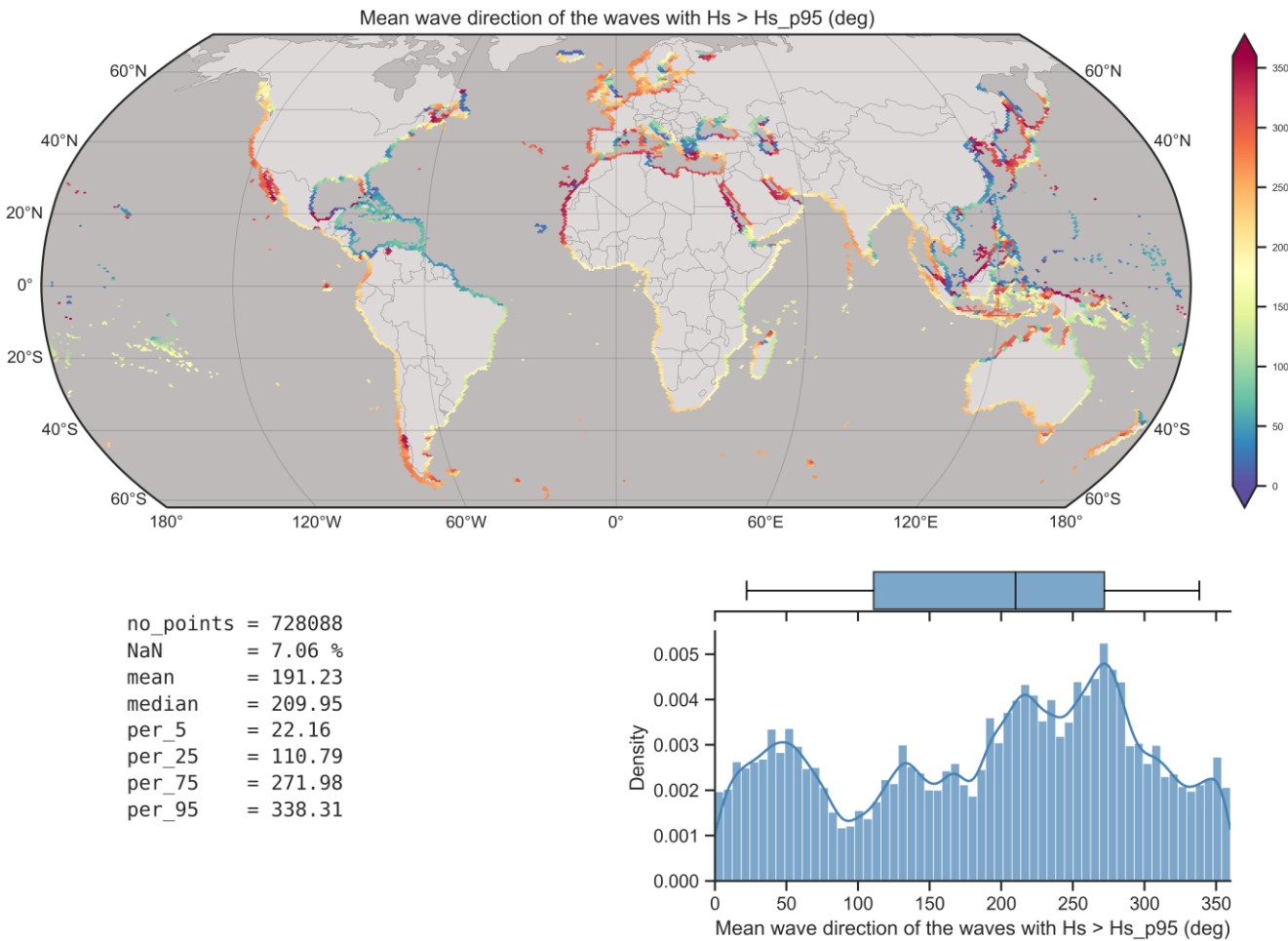

Figure 8: Global map and statistics of Average Mean Wave Direction (MWD) relative to true North, when Hs ≥ Hs_p95 (1979-2019). Top: Global map showing the median values in hexagons with a size of ~1 degree. Bottom right: Histogram and boxplot (with the lines showing the 5th and 95 percentiles, the box the 25th and 75th percentiles, and the vertical line the median) of Average Mean Wave Direction (MWD) at the transect level. Bottom left: various global statistics.

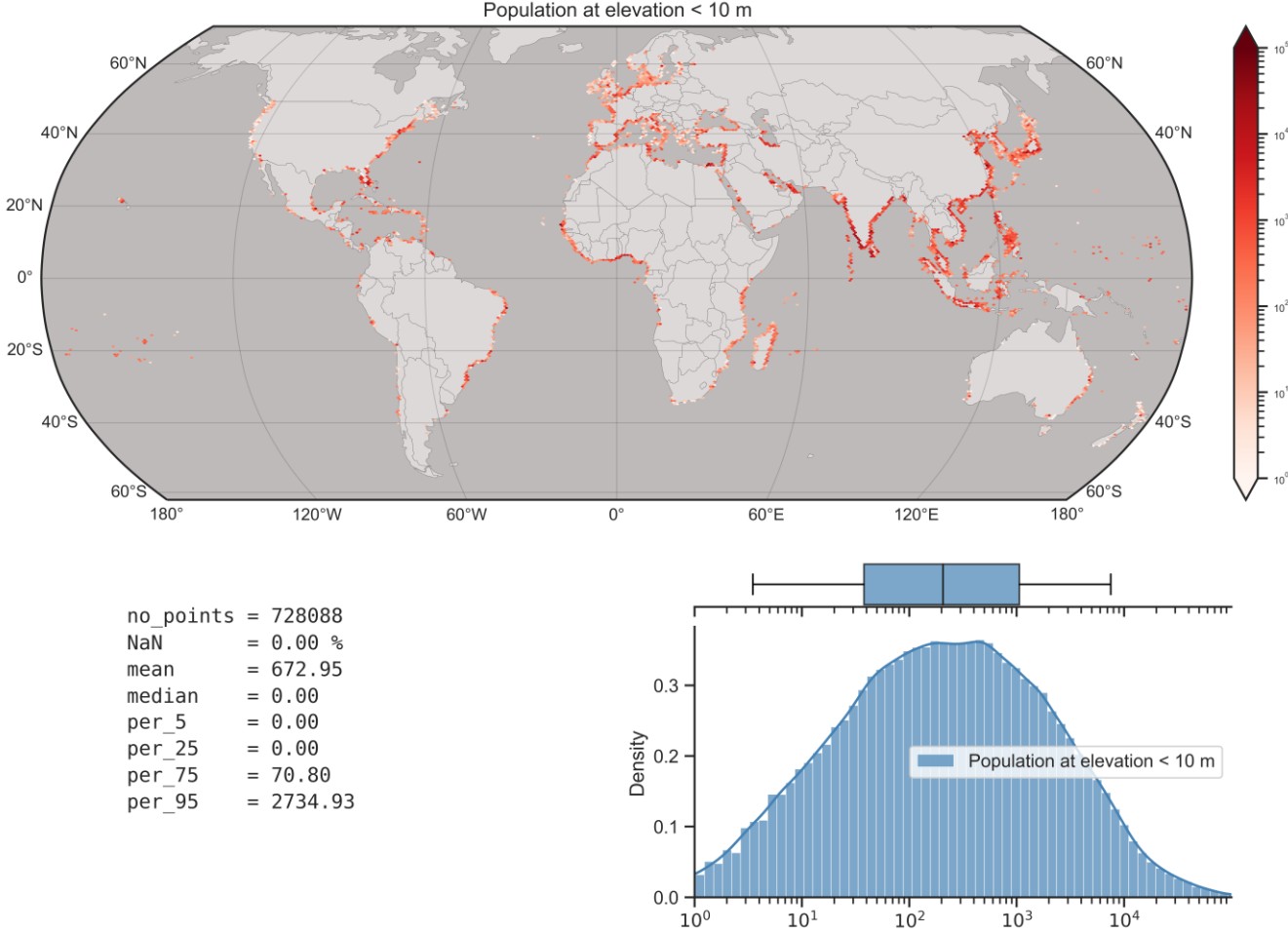

Figure 9: Global map and statistics of population at elevation < 10 m. Top: Global map showing the median values in hexagons with a size of ~1 degree. Hexagons with no population are not plotted. Colorbar is given in a logarithmic scale. Bottom right: Histogram and boxplot (with the lines showing the 5th and 95 percentiles, the box the 25th and 75th percentiles, and the vertical line the median) describing the transects with at least one person. The x axis is plotted in logarithmic scale. Bottom left: various global statistics (including transects with 0 population at elevation < 10 m).

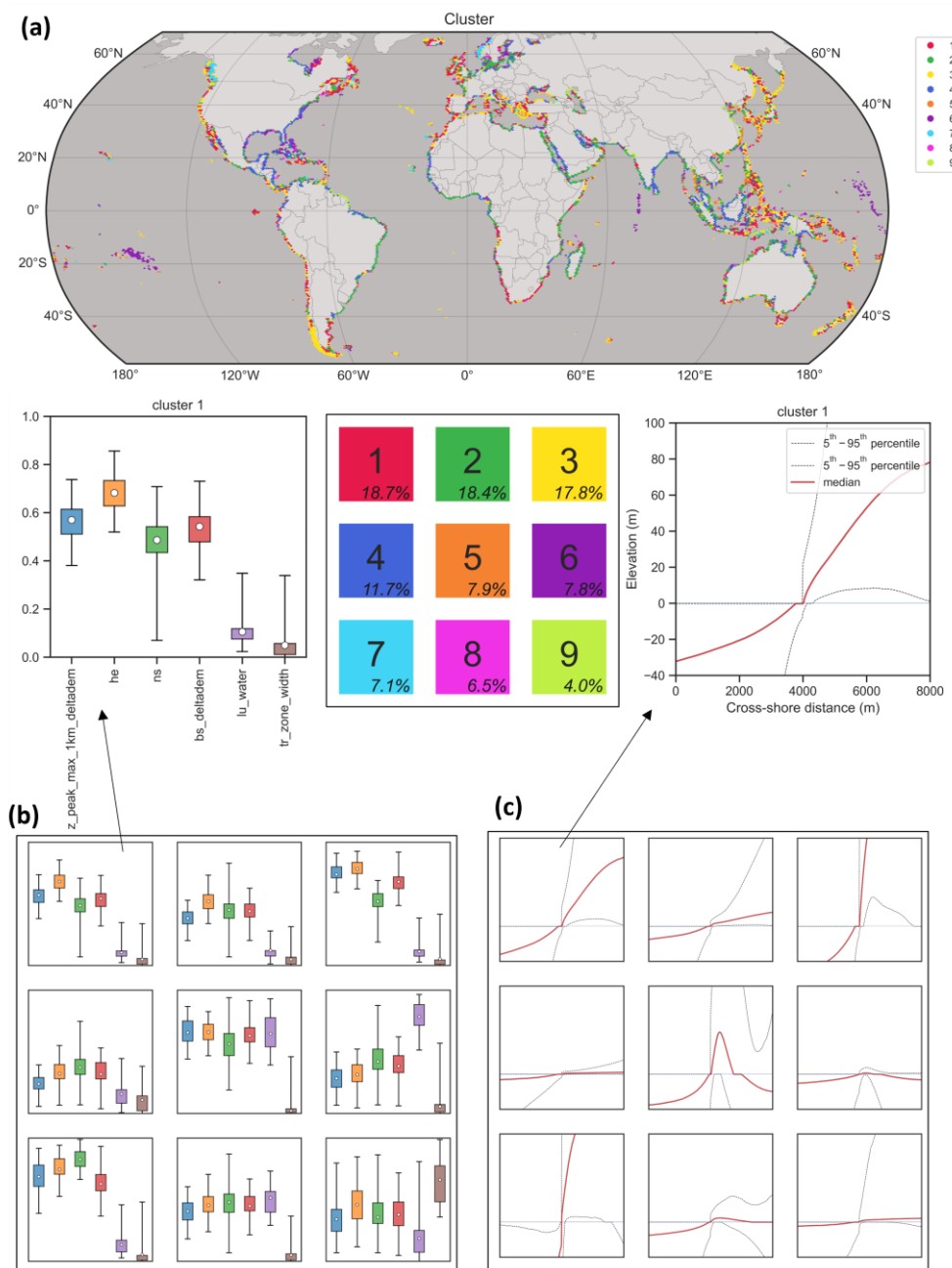

Figure 10: Geophysical clustering of the global transects with K-means into nine clusters using the *coastal Max (max peak 1 km)*, *mean hinterland elevation*, *nearshore slope*, *backshore slope*, *open-water land cover class occurrence and the transition zone width* as clustering parameters. (a) Global map showing the most frequent cluster in hexagons with a size of ~1 degree. The cluster number, frequency of appearance and color are shown in the grid in the middle, which is the same order as the grid at panels b and c. Cluster numbers are ordered by frequency of appearance. (b) Boxplots of intra-cluster scaled parameters variability. Boxes define the 25–75% percentile, lines the 5–95% percentile and circles indicate the centroid of each cluster. (c) Cluster elevation profile variability.