# Peer review of "Global Coastal Characteristics (GCC): A global dataset of geophysical, hydrodynamic, and socioeconomic coastal indicators"

_Earth System Science Data, 2023_

## Author Comment (AC1)

We would like to thank the four anonymous Reviewers for their comments and constructive suggestions. In this response we are addressing their comments and feedback. We have numbered the reviewer's comments (**R1.1, R1.2**... for Reviewer 1, **R2.1, R2.2** etc. for Reviewer 2 etc.) in order to facilitate referencing to each comment. We have added here all the changes in manuscript (shown with green). The pages and line numbers in our responses refer to the revised manuscript with track changes.

**Reviewer 1**

**This manuscript provided a global database of coastal characteristics on the geophysical, hydrometeorological and socioeconomic environment. This work is meaningful and the result is satisfactory. However, some other problems in the manuscript are still concerned in the following:**

**COMMENTS**

**R1.1:**
**More technical details on the database generation should be exposed.**

We have now tried to include more details on the database generation throughout the manuscript:

[Page: 4, Lines: 21-26]: "The choice of 1 km spacing was based on a balance between capturing information from high-resolution datasets (e.g., DEM and land use), while keeping the number of transects manageable from a data storage and computational point of view. This alongshore resolution offers a far better representation of the alongshore variability in comparison to previous studies where coarser resolution or segments approach were used (Wolff et al., 2016; Almar et al., 2021). Moreover, the transect system allows for future updates of the indicators based on new datasets that become available. The process adopted for creating the transects was as follows:..."

[Page: 5, Lines: 20-23]: "This process resulted in five profiles with 321 points with values from the 1) CopernicusDEM elevation, 2) DeltaDTM elevation, 3) GEBCO bathymetry, 4) CopernicusDEM water-mask, and 5) JRC water occurrence; and one profile with 801 points with values from the ESA World Cover land-cover for each transect. All these profiles were later used to derived transect-wise indicators (see Section 2.4)."

[Page: 5, Lines: 27-28]: "...(for the available land-cover classes in ESA World Cover please see Table 1)."

[Page: 6, Lines: 24-30]: "For characterizing the wave conditions at each transect, the ERA5 dataset was used. More specifically, the hourly time-series between 1979-2019 for the significant height of combined wind waves and swell (swh), the peak wave period (pp1d) and the mean wave direction (mwd) were used to extract specific indicators at the ERA5 grid locations (for more information on the parameters see (Hersbach et al., 2020)). These indicators included the 50th and 95th percentiles of the swh and pp1d, as indicators of the wave height and period during average and more extreme conditions. Additionally, the average mwd when the swh was larger than the 95th percentile of swh, was estimated as an indicator of the mean wave direction during extreme waves."

[Page: 7, Lines: 1-3]: "The ERA5 dataset was additionally used to extract indicators for mean daily temperature (t2m) and total daily precipitation (tp), for both of which the 50th and 95th percentiles were extracted from the hourly time-series between 1979-2019."

[Page: 7, Lines: 6-8]: "These included the mean higher high water (mhhw) and mean lower low water (mllw), 50th and 95th percentiles of SSL, and the SSL and TWL return values for 1, 2, 5, 10, 25, 50, 100 years (for more information on the generation of these indicators in GTSM_v3.0 please see Muis et al., 2023)."

[Page: 8, Lines: 20-24]: "Additionally, a warning/error flag was assigned to each transect based on the calculations explained above. These flags were: 0 for No errors/warnings , 1 when a shoreline point could not be found, 2 for when the DoC value was not available for that transect and the -10 m was used, 3 for when the DoC is deeper than the deepest profile point (which is used for the calculation), 4 for when Coastal Max (first peak) could not be found and 5 for when the nearshore slope was steeper than 1:5 and the transect was indicated as sandy."

**R1.2:**
**Could the authors show a flow chart of database generation?**

We have now added a flow chart in the start of the methods section.

[Page: 4, Lines 14-16]: "A general overview of the work-flow followed to create the GCC database is presented in  Figure 2. The generation of a global transect system and the methods to extract the various indicators from the datasets, based on these transects, are presented in more detail in the next sections."

[Figure]

Figure 2: Flowchart of the work-flow to derive the indicators of the GCC database.

**R1.3:**
**All the variables should be explained when they firstly appear in the text. Or else it will cause confusion.**

We have gone through the manuscript thoroughly to make sure that variables are explained when they first appear in the text.

**R1.4:**
**More analysis on the database is suggested.**

Due to the very high number of indicators (80) and the fact that a large part of them are based on other datasets, we felt that it would be an overwhelming amount of information to present figures similar to the 3 map figures that we have now in the results for all 80 indicators. For that reason we opted, for a single variable per variable group.

In case the reviewer was referring to the use of the dataset like the clustering example we currently present; we feel that since this is a Dataset description paper, it would be outside the scope of the paper to present an extensive analysis using the data, besides this representative classification that showcases the potential of the database. Besides, there are other studies that have already been published using this dataset (please see references below), and we are

confident that more studies in the future will make use of the GCC dataset to provide global 1[st] level assessments of coastal impacts, or more sophisticated classification of the coasts.

Vousdoukas, Michalis I., Panagiotis Athanasiou, Alessio Giardino, Lorenzo Mentaschi, Alessandro Stocchino, Robert E. Kopp, Pelayo Menéndez, Michael W. Beck, Roshanka Ranasinghe, and Luc Feyen. "Small Island Developing States under Threat by Rising Seas Even in a 1.5 °C Warming World." Nature Sustainability, October 9, 2023. https://doi.org/10.1038/s41893-023-01230-5.

Christiaanse, Jakob C., José A. A. Antolínez, Arjen P. Luijendijk, Panagiotis Athanasiou, Carlos M. Duarte, and Stefan Aarninkhof. "Distribution of Global Sea Turtle Nesting Explained from Regional-Scale Coastal Characteristics." Scientific Reports 14, no. 1 (January 8, 2024): 752. https://doi.org/10.1038/s41598-023-50239-5.

**Reviewer 2**

This paper develops a homogeneous diverse coastal dataset for about 80 diverse parameters around the world's coast at 1 km resolution. This is a good endeavour which I fully support. My comments below are designed to improve the manuscript.

**COMMENTS**

**Abstract**

**R2.1:**
**First three sentences do not flow to me – especially the "Therefore" – review and rephrase – what are you trying to say?**

We agree with the reviewer that this part of the abstract was not reading well, and have now changed it to:

[Page: 1, Lines: 12-15]: "More than 10 percent of the world's population live in coastal areas that are less than 10 meters above sea level (also known as the low elevation coastal zone – LECZ). These areas are of major importance for local economy, transport and are home to some of the richest ecosystems. At the same time, they are quite susceptible to extreme storms and sea level rise."

**R2.2:**
**'numerous open access global datasets' – roughly how many?**

It would be challenging to estimate the number of global datasets that are now becoming available. This is why we opted to use the word "numerous", to indicate the high number of datasets, but without making a guess. We prefer not to be speculative about this.

**Main Manuscript**

**R2.3:**
**Page 2, Line 15 to 29 – is just a list of papers – why these papers? For example, Kirezci et al. (2020) uses the DIVA segmentation that is earlier implied to be outdated. So I wonder when is higher resolution data required? For some questions, is this high resolution actually needed?**

We chose these papers since they are studies which, to our knowledge, deal with continental to global scale coastal impact assessments in the last decade. This list is not exhaustive. We added a closing sentence in the paragraph to highlight this:

[Page: 2-3, Lines: 32-2]: "The aforementioned studies are those that assess coastal impacts at large spatial scales using different data, grid resolutions and approaches, highlighting the need for consistent coastal data at the global scale."

In Kirezci et al. (2020) only the point locations from DIVA (which represent each of the DIVA segments) were used and not the actual data. Kirezci et al. (2020) used these locations in order to derive extreme water levels by combing newer data of tides, surges and sea level rise. However, the assessment of the flood extents in their study was performed on topography raster with a resolution of 1 km.

Indeed, we agree with the reviewer that a high resolution is not always needed. For example, with respect to the extreme water levels (especially since the underlaying global models have a course resolution), an alongshore resolution of 1 km would not have an added value. But, higher resolution does make a difference in, for instance, the profile shape characteristics for which the underlying DEM resolution is ~ 30 m.

We now added new text in the description of our transect creation giving more information on the choice of the alongshore spacing:

[Page: 4, Lines: 21-26]: "The choice of 1 km spacing was based on a balance between capturing information from high-resolution datasets (e.g., DEM and land use), while keeping the number of transects manageable from a data storage and computational point of view. This alongshore resolution offers a far better representation of the alongshore variability in comparison to previous studies where coarser resolution or segments approach were used (Wolff et al., 2016; Almar et al., 2021). Moreover, the transect system allows for future updates of the indicators based on new datasets that become available. The process adopted for creating the transects was as follows:"

**R2.4:**
**Page 3, Line 10 – why these datasets? As there are other possible datasets why select these ones?**

We recognize that there are more potential candidates for each type of global dataset. Nevertheless, we chose the ones used in this study, based on resolution, coverage and accuracy, which mostly was correlated with the time of publication. We mainly used the latest available datasets. To clarify this in the paper, we have now added this part:

[Page: 4, Lines: 9-11]: "While we recognize that there are more potential candidates for each type of dataset, we based our selection on resolution, accuracy and coverage. These indicators were largely connected with the date of publication. In general, we used the latest datasets that were available at the time of this study."

Additionally, after this initial manuscript was submitted a new coastal classification (Hulskamp et al., 2023) became available, so we decided to already update the sandy occurrence indicator with the output from that study which includes an update of the sandy location, in addition to extra classes (muddy, rocky, vegetated). To this end, we updated the related parts in the text:

[Page: 2, Lines: 30-32]: "In (Hulskamp et al., 2023) a combination of satellite imagery and geophysical data at a transect level was used to classify the global coastline to typologies, and the assess the dynamics of muddy coasts."

[Page: 6, Lines: 4-7]: "To assess the typology of the coast, the global coastal transect type classification from Hulskamp et al. (2023) was used. In that study, a different transect system (with 500 m alongshore spacing) was employed to classify coastal locations as sandy, muddy, rocky, vegetated or other type, using machine learning methods with satellite imagery and other geophysical indicators as input."

Hulskamp, R., Luijendijk, A., van Maren, B., Moreno-Rodenas, A., Calkoen, F., Kras, E., Lhermitte, S., and Aarninkhof, S.: Global distribution and dynamics of muddy coasts, Nat. Commun., 14, 8259, https://doi.org/10.1038/s41467-023-43819-6, 2023.

**R2.5:**
**Page 3, Line 24-26 – you say the data should not be used but if you have little money and data this might be better than nothing? So I would say recognise the great limitations if you do use this data. Of course the authors may disagree with me. I note later the authors discuss exploratory analyses, which is consistent with what I am saying.**

We agree with the comment from the reviewer and have now tried to adjust the text in order to not necessarily prohibit the use, but rather emphasize the need to understand the limitations:

[Page: 3-4, Lines: 28-2]: "However, it should be noted that, given the resolution of the parent data sets and underlying assumptions, the database presented here is not meant to be used for detailed local scale modelling or assessments. Therefore, the use of this dataset for this kind of assessments, for example at data poor locations, should necessarily take into account the underlying limitations and uncertainties."

**R2.6:**
**Page 6, Line 16 "The Depth of Closure (DoC) describes the depth seaward of which there is no significant change in bottom elevation" – for a specific timescale – later they have a timescale -- 34 years – but important the definition includes this concept.**

We have now added the reviewer's suggestion in the text:

[Page: 7, Lines: 22-23]: "The Depth of Closure (DoC) describes the depth seaward of which there is no significant change in bottom elevation at a specific timescale and is determined by the wave statistics (Hallermeier, 1978)."

**R2.7:**
**Page 6, Line 20 "A limit of 150 km was used" – don't understand?**

This limit refers to the buffer zone around each of our transects that we looked for points from Athanasiou et al. 2019, in order to get an estimate of the local DoC. We have now tried to clarify this in the text:

[Page: 7, Lines: 27-28]: "A buffer zone of 150 km around each transect was used to sample the offshore points, to avoid values that are non-representative of the offshore wave environments of the transects being ascribed to them."

**R2.8:**
**Page 8 Line 3, "12.11 %" – 12.1 % or even just 12 %? -- in general report numbers to an appropriate precision.**

We agree with the reviewer that this precision is meaningless in this case, so we updated this results to:

[Page: 9, Lines: 14-15]: "For ~12% of the global transects, a coastal max with this method could not be calculated, since there was no elevation peak in the 1 km zone."

**R2.9:**
**Page 8, Line 23 – agree here.**

We are happy that this part is in line with the reviewer's point of view.

**R2.10:**
**Page 8 Line 10 –earlier analyses have emphasised that while the coast contains a lot of people, much of world's coast has little population so this makes sense.**

We are happy that our results are in line with previous analysis.

**R2.11:**
**Page 10, line 14 – 'past years' – vague – the authors seem to be looking at the last 5 to 10 years? Define time scale.**

We have now made the time scale more explicit by changing to:

[Page: 12, Line: 7] **: "**During the last decade or so, various studies have assessed…**"**

**Reviewer 3**

**The manuscript presents a global database of coastal attributes, which includes data on 80 indicators of the physical and socioeconomic characteristics of the global coast based on freely available global datasets. This information is provided for over 700,000 points of the ice-free part of the global coastline. The work follows up on, and extends, past efforts and provides a new and updated view of the global coastal environment, using a data model that is based on cross-shore transects. The manuscript is well written, interesting and presents clearly the main aspects of the work. The manuscript is suitable for publication – nevertheless, I have a few questions that I believe require some clarification; a series of recommendations that the authors may find useful for improving the manuscript; and a couple of comments of editorial nature. I have listed those below:**

**R3.1:**
**Previous work has been based on coastline segmentation (and segments) – what are the benefits/advantages of using the transect system? Although this system also encapsulates some sort of segmentation, it would be useful to briefly discuss whether and how this is an advancement or simply a viable alternative.**

Previous work e.g. DIVA, used coastal segments with quite large individual segments (DIVA included 12,148 coastal segments globally, versus the almost 700,000 transects we present here). The segmentation in DIVA was performed by combining a set of physical, administrative and socio-economic criteria based on available data at the time, to create segments with coherent characteristics. However, this low number of segments at the global scale would still mean that a big part of the coastal variability is lost by this aggregation. Additionally, the segmentation is strongly biased towards the various data that are used, meaning that updating the underlaying data, could potentially result in a quite different segment system.

While the location of the coastline is dynamic as well, we believe that it is a more coherent reference system, which allows us to update the characteristics based on new datasets becoming available. Additionally, a segment system will not allow for a structured capture of coastal maxima at the desired resolution. Last but not list, we believe that our transect system does not replace a segment system but complements it. For example, our data could be aggregated in segment systems to calculate descriptive statistics without defining the segments sizes a priori.

We have now added this part in section 2.1, to justify our choice:

[Page: 4, Lines: 21-25]: "The choice of 1 km spacing was based on a balance between capturing information from high-resolution datasets (e.g., DEM and land use), while keeping the number of transects manageable from a data storage and computational point of view. This alongshore resolution offers a far better representation of the alongshore variability in comparison to previous studies where coarser resolution or segments approach were used (Wolff et al., 2016; Almar et al., 2021). Moreover, the transect system allows for future updates of the indicators based on new datasets that become available."

**R3.2:**
**If I understand correctly, the authors have calculated zonal statistics to estimate the indicators and mention that these buffers can overlap. Wouldn't this imply that information is being considered twice in the zonal statistics in those cases where the buffers overlap? And isn't this a problem? I understand that the authors discuss this in the case of**

**population, but the problem should show up elsewhere and I believe that it should be technically possible to somehow avoid this double counting.**

The reviewer is correct that in some cases this can lead to information been considered twice in the zonal statistics. We describe this in the limitations sections of the discussion:

[Page: 11, Lines: 4-6]: "…For example, when the coast has a convex shape (e.g., at a peninsula) the transects and thus buffer zones can overlap, meaning that the same cell of population or land-use from the initial raster datasets can be counted in more than one transect…"

We calculate zonal statistics only for the population and land-use indicators (please see newly added Figure 2), that is why we mention only these two in the discussion section. While we understand the suggestion from the reviewer, in this work we focus on providing descriptive indicators of the local conditions at different locations along the coast. For example, in the case of a narrow island, and two transects from different sides of the island that overlap, the population might indeed be counted twice. However, it will still mean that this population is exposed to coastal flooding from either side of the island.

We now added an extra part in the text to highlight this:

[Page: 11, Lines: 9-11]: "The purpose of the population and land-cover indicators is to provide an indication of the exposure and the socioeconomic characteristics of the area around the transect. To that end, these indicators should not be used to calculate aggregated summed values e.g., to estimate the total population near the coastline."

**R3.3:**
**On the buffers again – do these extend up to 4km inland? If this is actually the case, isn't this strip too narrow to describe coastal features for certain processes? For example, although in certain regions floodplains are indeed narrow in general (e.g. Mediterranean), there are places where floodplains can extend tens or hundreds of kilometers inland (e.g. south Asia).**

Our transects indeed extend with a fixed length of 4 km inland. Certainly, this would mean that really flat floodplains are not captured at their full extent, but since our main focus was to capture the characteristics close to the coastline with respect to coastal maxima, slopes etc. and not to perform inundation modelling we believe that this is a reasonable methodological choice for a global study. Moreover, having uniform lengths for our transects elevation profiles was important for data storage.

Nevertheless, this was not mentioned explicitly in the text before, therefore, we have added a part now in the methods sections:

[Page: 5, Lines: 6-8]: "While the 4 km length landwards might now always cover the full extents of coastal floodplains at flat coastal areas, this choice was deemed appropriate, since the focus of the derived dataset is to capture the coastal characteristics close to the coastline and not to perform coastal inundation modelling."

**R3.4:**
**How do authors treat mismatches between datasets? For example, the selected coastline and e.g. population dataset may not have matching coastlines with the result that when overlaying those, population may appear to "live in the water"; similar situations may**

**occur with other datasets. How are these issues accounted for?**

Indeed, the high number of datasets and the different sources do not guarantee a perfect match on the transition between land and water. To account for this in the context of the elevation profiles we have used the ESA World cover map to classify sea and land. Moreover, on the population issue, the WorldPop data that we use are constrained using global building footprint information which ensures to a large extent that people are not located in the water.

We realized that we had not explicitly mentioned that the WorldPop dataset that we used was the constrained one in previous version of the manuscript, so we have added that now in Table 1, at the WorldPop description:

"Global population count per pixel at ~100 m resolution (Constrained individual countries 2020 UN adjusted)"

**R3.5:**
**Maybe the authors could consider including population projections in their database as it seems to me that this could be useful and not too demanding based on how they processed the data. Future population spatial datasets exist for all SSP (e.g. Jones and O'Neill, 2016, or for coastal population Merkens et al., 2016).**

We appreciate the reviewer's suggestion regarding the inclusion of population projections from available future gridded population datasets. While we acknowledge the potential utility of such indicators, we opted not to include these in our current analysis for several reasons.

Firstly, the proposed dataset have a quite coarse resolution ranging from ~1 km (Jones and O'Neill, 2016) to ~14 km (Merkens et al., 2016). This means that it would be quite difficult to sample these datasets at the quite finer spatial extents of our transects buffer zones.

Additionally, our study focuses on current characteristics and including future projections for population would potentially mean that for consistency, future projections of sea level rise, wave characteristics etc. should be included as well. This would broaden the scope beyond our research objectives. In addition, such projections are updated quite frequently in view of new scenarios, while baseline datasets will change only if there are new (better) data available.

**R3.6:**
**In those cases where proximity was used to assign data, how did the authors address the overlapping of classes? (e.g. a sandy point from 1 km away does not necessarily lead to a sandy transect but possibly to a mixed morphology one).**

During this revision we have used the newly published dataset of Hulskamp et al., 2023 (please also see our answer to Reviewer 2 at R2.4) to replace the previous indicator of the presence of sandy beaches to a more general coastal classification indicator which included the classes: sandy, muddy, rocky and vegetated. To this end, we classify our transects using their points and the proximity of 1 km. In case there are more than one points in the buffer zone, we assume that the closest point is the most appropriate.

We now describe this in the text:

[Page: 6, Lines: 7-10]: "To classify our transects by a coastal type, proximity analysis was used (see Figure 5a) to determine the presence of points from Hulskamp et al. (2023) in a 1 km buffer

zone around the centroid of our transects. If more than one points were present in the buffer zone, the closest point was selected."

**R3.7:**
**Nowadays, when there is constant production of new data, the database can fairly quickly become obsolete (at least regarding some of the indicators). Is there a plan to provide data updates? Also, with respect to the static coastline, is it technically feasible to update the coastline without repeating the entire work?**

We fully acknowledge the reviewer's concern on the future relevance of our dataset when new global datasets become available. The way we have set up our workflow will allow us to update the indicators in the future when datasets that are deemed more accurate become available. Moreover, new indicators could be added in future versions. As an example, we already updated the coastal classification indicator (see comment R3.6) for this manuscript revision.

Furthermore, we have uploaded the dataset in Zenodo with a specific version identifier (v1.0 for the initial submission and v1.1 for the changes of this revision). This version control allows for keeping track of all future changes.

Moreover we highlight the potential updates of the database in the conclusion section:

[Page: 13, Lines: 11-13]: "The indicators were extracted based on various open-source global datasets that were available at the time of undertaking this work, but can be updated using the same workflow, whenever new datasets become available."

**R3.8:**
**The authors could maybe discuss the issue of consistency, which is not trivial in global analysis and an asset of their work. Their database provides consistent (and therefore comparable) information for the entire globe, which can be a significant advantage in global assessments.**

We thank the reviewer for highlighting the advantages of our dataset. We have now added this part in the conclusions:

[Page: 13, Lines: 8-10]: "We believe that our dataset can act as a significant asset for future global assessments of coastal hazards and impacts, since it provides information in a consistent and therefore comparable manner."

**R3.9:**
**An important (I believe) comment: the authors provide a simple metadata file, with very limited information on the included parameters. Metadata are in many cases as important as the data themselves and provide significant added value to databases. The current paper, although detailed and complete, does not really present detailed information on e.g. technical issues related to processing and respective decisions, in order for the work to be able to replicate. I would therefore recommend the authors to extend the metadata document (possibly following existing metadata standards for geospatial data) to accompany the database. Although I am aware that this work is tedious and time consuming, I can only emphasise how important it is to their database being extensively used in the future.**

We thank the reviewer for his/her suggestion and fully agree on the importance of metadata accompanying datasets. However we believe that this paper is an integral part of the presented dataset and including detailed information on the methodology (which already is presented in the paper) in the metadata file will diminish the purpose of having a data description paper in the first place. Moreover, we believe that from a practical point of view, it is far easier to follow the methodology and work done in a paper format (with sections, tables and figures), rather than a metadata file format. In zenodo, where the dataset is hosted, there is "Is described by" section where we already connect with the paper pre-print in an explicit manner.

We believe that a user could use the current metadata file as a look-up table to interpret the csv files and understand the provided indicators (e.g., full names, units and a quick description on how they were derived), while the user could turn to the accompanying data description paper for detailed information on the methodology and the general workflows used when needed.

**R3.10:**
**Further to the previous comment, shouldn't the authors also provide the transects used for the compilation of the databse?**

We have decided not to provide the transects as a separates spatial file mainly for minimizing the dataset size. Nevertheless, since we provide the location (lat, lon) and angle of the transect as indicators, someone could infer the actual transect by simple geometrical procedures.

**R3.11:**
**A minor comment:line 30 reads as if population makes the coast one of the most valuable ecosystems, I would suggest to rephrase.**

We agree that this part was in need of rephrasing to make the meaning clear. We have now changed to:

[Page: 1, Lines: 28-30]: "Between 750 million and 1.1 billion people live in low-lying coastal areas (MacManus et al., 2021) which is expected to increase in the future (Neumann et al., 2015). This highlights the considerable economic significance of coastal regions, which also represent some of the planet's most valuable ecosystems (Paprotny et al., 2021)"

**Reviewer 4**

**The manuscript presents a new compilation of indicators that characterize the global ice-free coastal areas. The paper effectively presents technical details for each of the indicators and how they were integrated into the database, which appears to be an advance over existing datasets. A useful discussion of limitations is included, as well as a helpful example of use of the dataset for the application of coastal classification.**

**Overall, the presentation is of good quality, but there are some questions that need attention and some technical corrections to be made.**

**R4.1:**
**Why was the global 10-m land cover dataset chosen? Is it simply because it is the highest resolution global land cover data, or the most recent? Or, is it the best accuracy? There are multiple other global land cover products in the 30-m to 100-m spatial resolution range. How does the selected 10-m data compare in accuracy to the other global land cover products?**

We appreciate the comment from the reviewer and we realize we did not go in depth in the manuscript on how we made this decision. Since, there is plethora of available product for the different variables that we wanted to map, we had to make some decisions based on date of publication, accuracy and resolution. We briefly mention this in the introduction of Section 2:

[Page: 4, Lines: 2-5]: "The prerequisites for all data sets used here to extract coastal indicators was that the datasets were recent open-access and had a global coverage. While we recognize that there are more potential candidates for each type of dataset, we based our selection on resolution, accuracy and coverage. These indicators were largely connected with the date of publication. In general, we used the latest datasets that were available at the time of this study."

Our choice to use ESA world cover was based on its high-resolution of 10 m and high accuracy of 77%. We added this part in the text:

[Page: 4, Lines: 12-14]: "For example, for land cover information we used the ESA World Cover map, since it had the highest resolution of global land cover products available at the time and a high accuracy of 77% (Zanaga et al., 2021)."

**R4.2:**
**The discussion in Section 2.1 ('Transect system') implies "shore normal" transects, but it doesn't explicitly state that. Is it true that the transects are shore normal? If so, how was that specifically done, the orientation of the transect to the shoreline? More information is needed here.**

We apologize if this was not clear in the manuscript before. We have now tried to clarify this, by adding this part in section 2.1:

[Page: 5, Lines: 1-2]: "Then using the spacing of 1 km, shore normal transects were created by using the local orientation of the coast at each 1 km interval."

**R4.3:**
**"DeltaDEM" is used throughout the paper, but when going to the citation for the dataset (https://doi.org/10.4121/21997565) it is found that the dataset name is actually**

**"DeltaDTM". Why the discrepancy? This is confusing, and should be rectified.**

We thank the reviewer for pointing this discrepancy out. We have now corrected this throughout the document and the dataset.

**Specific Comments:**

**R4.4:**
**Page 3, lines 1-3: See also DiluviumDEM (Dusseau et al., 2023) for a new corrected DEM for coastal areas**

DiluviumDEM was not yet published when we initially submitted our manuscript. We have now added a reference to this DEM when we talk about the available datasets:

[Page: 3, Lines: 6-8]: "In an attempt to correct for these effects, new DEMs have been produced correcting the elevations in vegetated and urbanized areas, such as FABDEM (Hawker et al., 2022), which is however not in the public domain, and more recently the open access DiluviumDEM (Dusseau et al., 2023) and DeltaDTM (Pronk et al., 2024)."

**R4.5:**
**Page 3, line 17: "1 km interval" – Why was a 1-km interval chosen? Is this a big improvement over DIVA? More information is needed here on the rationale for choosing the 1-km interval between transects.**

We have added justification of this choice in the methods sections (see also our response to Reviewer 3, in R3.1):

[Page: 4, Lines: 21-25]: "The choice of 1 km spacing was based on a balance between capturing information from high-resolution datasets (e.g., DEM and land use), while keeping the number of transects manageable from a data storage and computational point of view. This alongshore resolution offers a far better representation of the alongshore variability in comparison to previous studies where coarser resolution or segments approach were used (Wolff et al., 2016; Almar et al., 2021). Moreover, the transect system allows for future updates of the indicators based on new datasets that become available."

**R4.6:**
**Page 4, line 9: "a simple smoothing procedure" – Is there a reference for this? Or the name of the function in a software package? More information is needed, especially if the procedure is to be replicated.**

We have now added a part clarifying the algorithm and software used:

[Page: 4, Lines: 27-29]: "The smoothing procedure was conducted using Chaiken's algorithm as implemented in QGIS (QGIS Development Team, 2023) using a 0.25 offset, 180 maximum node angle and 5 iterations."

**R4.7:**
**Page 4, line 15: "coastal zone" – How is the coastal zone defined? Less than 10 m in elevation? Please specify.**

Here by coastal zone we mean the area that included the first coastal features like dunes, dikes

etc. We have now added a part to more clearly clarify the choice of the 4 km (please also see comment R3.3):

[Page: 5, Lines: 6-8]: "While the 4 km length landwards might not always cover the full extents of coastal floodplains at flat coastal areas, this choice was deemed appropriate, since the focus of the derived dataset is to capture the coastal characteristics close to the coastline and not to perform coastal inundation modelling."

**R4.8:**
**Page 5, lines 3-4: "the number of people located below specific elevation thresholds (1,5 and 10 m above MSL)" – It is likely that the DEMs used (CopernicusDEM and DeltaDEM) are not accurate to the 1-m level, so slicing elevation at 1-m would have much uncertainty. Delineating the 1-m elevation zone is analogous to drawing a contour line at 1-m, which would require the elevation data to be much more accurate for a high-confidence 1-m contour (see Gesch, 2023, https://doi.org/ 10.5281/zenodo.8011577). At the least, this limitation of using global DEMs for delineation of a 1-m elevation zone should be noted.**

We thank the reviewer for pointing this out. We agree that the uncertainty can be high due to the lack of accuracy at the 1 m level of the global DEMs. We have now added the following lines in the limitation section of the discussion:

[Page: 11, Lines: 12-15]: "While we provide indicators of population counts at different elevation thresholds, it should be noted that the uncertainties in the derived values can be high, since they are dependent on the accuracy of the global elevation model used. In the case of the present study, we used DeltaDTM which has a vertical mean absolute error of 0.45 m overall (Pronk et al., 2024), which gave us confidence on the derived indicators giving a good approximation of the exposed population."

**R4.9:**
**Page 6, lines 7-8: "CopernicusDEM/DeltaDEM topography profile values were used for the land cell elevations" – It is unclear what this means exactly. Was DeltaDEM used for 0-10 m in elevation along each transect and the CopernicusDEM used for elevations greater than 10 m? More explanation is needed here.**

As we describe in the beginning of the paragraph, we create two different elevation profiles, one based on CopernicusDEM and one on DeltaDTM. To this end, we produce two variation of the parameters that are associated with coastal maxima and slopes. We now did a small update in the text in hope that this reads better now:

[Page: 7, Lines: 11-15]: "Two elevation profiles were created per transect, one using the CopernicusDEM and one using the DeltaDTM as the topography source. The following steps were followed : 1) The CopernicusDEM mask profile was used to define the land cells, 2) The CopernicusDEM or DeltaDTM topography profile values were used for the land cell elevations after they were transformed from "m above geoid" to "m above MSL" using the mean dynamic topography map (DTU10_MDT), the value of which was saved in the database as well,…"

We further describe the derivation of the double indicators in the methodological section 2.4 (Page 8, Lines 7-9) and in the discussion section 4.1 (Page: 10, Lines: 27-30)

**R4.10:**
**Page 7, line 10: "overcorrections" – How is overcorrection determined? More information**

**is needed here.**

This connects with the previous comment (R4.9). Essentially, we discuss what this overcorrection means in the discussion section 4.1 (Page: 10, Lines: 27-30).

**Technical Corrections:**

**R4.11:**
**Page 2, line 27: "relative course" should be "relatively coarse"**

We have now corrected this.

**R4.12:**
**Page 5, 23: "GTSM" – Define the abbreviation here, as it is the first time used in the body of the paper.**

We have now added the full name here:

[Page: 6, Line: 19]: "the Global Tide and Surge Model (GTSM_v3.0)"

**R4.13:**
**Page 6, line 23: "10 m" – should this be "-10 m"?**

This value describes the Depth of Closure, which is a positive value since it is a depth.

**R4.14:**
**Page 8, line 7: "MWD" -- Define the abbreviation here, as it is the first time used in the body of the paper.**

We refer to Table 3 in the previous sentence, in which we define this abbreviation. To this end we believe that we do not need to define this here as well.

**R4.15:**
**Page 13, lines 13-14: There is no journal identified in this reference.**

We have now added the journal:

"Athanasiou, P., van Dongeren, A., Giardino, A., Vousdoukas, M., Gaytan-Aguilar, S., and Ranasinghe, R.: Global distribution of nearshore slopes with implications for coastal retreat, Earth Syst. Sci. Data, 11, 1515–1529, https://doi.org/10.5194/essd-11-1515-2019, 2019. "

**R4.16:**
**Page 14, lines 31-32: There is no journal identified in this reference. Also, why is the title all in capital letters?**

We have now corrected this citation:

"Knudsen, P. and Anderson, O. B.: A Global Mean Ocean Circulation Estimation Using GOCE Gravity Models- The DTU12MDT Mean Dynamic Topography Model, in: 20 Years of Progress in Radar Altimatry, 20 Years of Progress in Radar Altimatry, Venice, Italy, ADS Bibcode: 2013ESASP.710E.130K, 130, 2013."

---

## Author Response (AR2)

We would like to thank the two anonymous reviewers (1 and 2) for accepting the revised version of our manuscript. Here we only address the minor comment from anonymous reviewer 2.

**Reviewer 2**

**The authors have revised the paper well and I am happy with the improved manuscript. This paper is of high quality and I commend the authors for their effort -- this will support a range of coastal assessments.**

We appreciate the positive comments of reviewer 2.

**I have one small correction in Section 3.3. '1 person' should be 'one person' -- there are two cases that need changing -- once this is done the editor can accept.**

[Page 9, Lines 25-2]: "…, while for the transects that have at least one person, the median value is ~200 people, with a distribution that is close to a normal distribution in the logarithmic scale. The 25th and 75th percentile values are 40 and 1000 people respectively for the transects with at least one person."